# GeoPE: A Unified Geometric Positional Embedding for Structured Tensors

## Abstract

Standard Vision Transformers flatten 2D images into 1D sequences, disrupting the natural spatial topology. While Rotary Positional Embedding (RoPE) excels in 1D, it inherits this limitation, often treating spatially distant patches (e.g., at row edges) as sequence neighbors. Existing 2D approaches typically treat spatial axes independently, failing to decouple this false sequential proximity from true spatial distance. To restore the 2D spatial manifold, we introduce Geometric Positional Embedding (GeoPE), a framework that extends rotations to 3D Euclidean space using quaternions. To overcome non-commutativity and ensure symmetry, GeoPE constructs a unified rotational operator by computing the geometric mean in the Lie algebra. This creates a geometrically coupled encoding that effectively separates spatial dimensions. Extensive experiments on image classification, object detection, and 3D semantic segmentation demonstrate that GeoPE consistently outperforms existing 2D RoPE variants and significantly enhances shape bias, confirming its ability to capture true geometric structure.

## 1 Introduction

Transformer (Vaswani et al., 2017) has emerged as the backbone of large language models due to its capacity to capture global dependencies and generalize across modalities. However, Transformer lacks an inherent mechanism for sequence order (Devlin et al., 2019; Raffel et al., 2020; Shaw et al., 2018). Conventional positional encodings like Absolute Positional Encodings (APE) (Devlin et al., 2019; Chen et al., 2021) and Relative Positional Encodings (RPE) (Liu et al., 2021; Park et al., 2022; Wu et al., 2021) inject position information but often face trade-offs between flexibility and complexity. Rotary Positional Encoding (RoPE) (Su et al., 2024) overcomes these limitations by rotating query and key vectors in a 2D plane, providing attention with strong length generalization (Jiang et al., 2023; Touvron et al., 2023; Yao, 2024).

With Transformer increasingly applied to vision tasks, researchers have explored extending RoPE to two dimensions (Fang et al., 2024; Lu et al., 2024a;b). However, standard Vision Transformers (ViT) (Dosovitskiy et al., 2020) process images by flattening 2D grids into 1D sequences. This operation creates a geometric mismatch where spatially distant patches (e.g., at row edges) become immediate sequence neighbors. Existing 2D methods often adopt axis-wise designs, processing horizontal and vertical encodings independently or via mixed frequencies (Chu et al., 2024). For instance, Heo et al. (2024) partitions the embedding space to allow independent rotations per axis. Nevertheless, because these axes are not geometrically coupled, such approaches struggle to decouple the false sequential proximity created by flattening from true spatial locality, effectively leaving the weak cross-axis interaction of high-dimensional RoPEs unresolved.

The challenge of modeling this coupling is amplified in multi-modal learning (Dao et al., 2024; Yin et al., 2025; Shu et al., 2023). Some works extend RoPE to higher dimensions via Lie group/algebra frameworks (Appendix B). For example, Liu & Zhou (2025) formalizes RoPE using a maximal abelian subalgebra (MASA) and introduces cross-dimensional interactions through orthogonal basis changes. However, this can overly constrain representations or incur high computational costs. Comminiello et al. (2024) argues that hypercomplex algebras provide essential inductive biases for multidimensional structures. Alternatively, Ostmeier et al. learn dense skew-symmetric matrices to build rotation operators, yet this remains computationally expensive and lacks theoretical guarantees for efficient spatial reconstruction.

We propose Geometric Positional Embedding (GeoPE), which extends RoPE's 2D complex-plane rotations to 3D Euclidean space using quaternions to strictly model coupled rotations in structured tensors (Section 3.3). Unlike independent axial methods, GeoPE treats spatial dimensions as a unified geometric entity. To overcome the non-commutativity of quaternion multiplication and ensure a consistent spatial prior, we construct a unified rotational operator by computing the symmetric mean in the logarithmic tangent space (Section 3.2). We also propose a linear variant for direct relative encoding (Section 3.4). This method enriches self-attention with a geometrically meaningful understanding of space, thereby fostering superior spatial reasoning and shape awareness (Section 4). Experiments (Section 5) show that GeoPE achieves significant performance gains in classification, detection, and segmentation, while retaining strong extrapolation properties.

## 2 RELATED WORK

**Position Encodings**. Transformers lack inherent positional awareness and thus rely on encodings to capture the order of tokens. The original Transformer (Vaswani et al., 2017) employs sinusoidal absolute positional encodings (APE), which generalize poorly to long sequences. In contrast, learnable APE (Shaw et al., 2018) improves flexibility and representation for tasks such as sentence alignment and context modeling. Vision Transformers (ViT) (Dosovitskiy et al., 2020) similarly adopt learnable APE for image patches. Relative positional encodings (RPE) model pairwise token distances, supporting long sequences and cross-sequence dependencies (Liu et al., 2021; Shaw et al., 2018), though naive designs incur quadratic cost. Rotary Positional Encoding (RoPE) (Su et al., 2024) encodes relative positions via complex-plane rotations and is widely used in large language models; however, its performance degrades when extrapolated to much longer contexts. More recent approaches learn semanticized position structures. Contextual positional encodings (CoPE) (Golovneva et al., 2024) enhance reasoning and mathematical capabilities. Abacus embeddings (McLeish et al., 2024) capture numerical structures for arithmetic, while lightweight methods, such as LaPE (Yu et al., 2023), apply adaptive normalization to improve robustness across architectures.

**RoPE in Visual Model**. RoPE has demonstrated strong extrapolation capabilities in long-text modeling and dialogue, motivating its extension to vision and multimodal tasks (Lu et al., 2024b; Wang et al., 2024; Yao et al., 2024). A straightforward adaptation applies 1D RoPE to ViT variants, as in Hybrid X-former (Jeevan & Sethi, 2022). However, gains are modest and have been validated only on small datasets (e.g., CIFAR, Tiny ImageNet). To better handle 2D inputs, works such as EVA-02 (Fang et al., 2024) and Unified-IO 2 (Lu et al., 2024a) have incorporated axial 2D RoPE into multimodal and diffusion models; however, these fail to capture diagonal interactions. RoPE for ViT (Heo et al., 2024) further proposed RoPE-Mixed, which combines axial frequencies to enhance 2D encodings and downstream performance. However, this approach remains essentially frequency composition, offering only loose dimensional coupling and limited generality. Qin et al. (2023) proposes a Quaternion Product Unit (QPU) that leverages quaternion algebra and the laws of the $3D$ rotation group ($SO(3)$). By representing $3D$ rotation data as quaternions, their work demonstrates that complex algebras can effectively maintain geometric structure and achieve superior robustness in rotation-sensitive tasks, which strongly aligns with the geometric approach of GeoPE.

**Shape Bias**. Cognitive science has shown that humans rely primarily on global shape, rather than texture or color, for object recognition and lexical learning, whereas CNNs exhibit a different tendency. Hosseini et al. (2018) demonstrated that standard CNNs often lack shape bias, instead depending heavily on local texture or color cues. However, Ritter et al. (2017) reported that networks can develop shape preference under certain conditions. To examine this systematically, Geirhos et al. (2018) compared CNNs and humans using style-transferred images with conflicting shape and texture information. While humans consistently prioritize shape, CNNs tend to favor texture. To mitigate this bias, they introduced Stylized-ImageNet, which reduced texture reliance and induced stronger shape bias, yielding models with improved robustness and transferability. These findings suggest that enhancing shape bias can make models more human-like while also strengthening generalization.

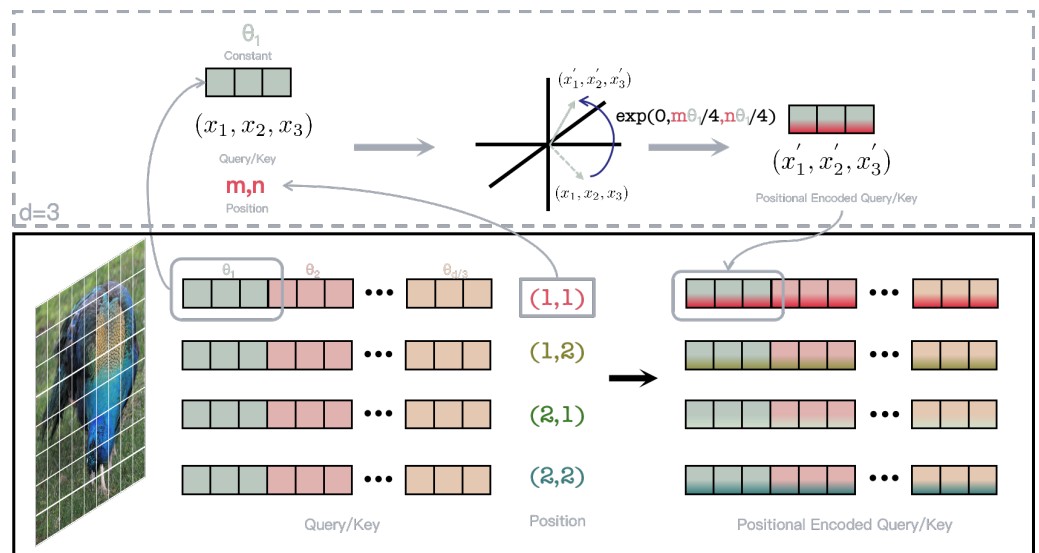

Figure 1: **Geometric Transform of Geometric Positional Embedding (GeoPE).** This figure illustrates how GeoPE encodes $2D$ positions (e.g., $(m, n)$) by extending Rotary Positional Embedding (RoPE) to $3D$ space using **quaternions**. For each feature sub-vector $(x_1, x_2, x_3)$, GeoPE calculates the **geometric mean** of the height and width rotations in the **Lie algebra** to create a unified, symmetric rotation operator. This operator then applies a geometrically coupled $3D$ rotation to the query/key sub-vector via a **sandwich product** ($\mathbf{p}' = \mathbf{rpr}^*$) to inject the positional bias.

## 3 METHODOLOGY

In this section, we detail the formulation and implementation of GeoPE. We first establish the geometric requirements for multi-axial rotation in Section 3.1, then construct a symmetric rotational operator using Lie theory in Section 3.2. Finally, we demonstrate the framework's extension to 3D in Section 3.3 and propose a linear variant in Section 3.4.

### 3.1 GENERALIZING ROTATIONS TO 3D SPACE

While RoPE effectively models 1D sequence distance introduced by Appendix H, it cannot distinguish between the 'sequence neighbors' created by flattening and true 'spatial neighbors.' To resolve this ambiguity, we extend the rotational domain to 3D Euclidean spaceas illustrated in Figure 1 using quaternions (introduced in Appendix A). By mapping height and width to orthogonal rotational axes (using $j$ and $k$ components), we ensure that sequence-adjacent but spatially-distant patches induce drastically different rotational states, effectively recovering the 2D manifold.

Mathematically, a feature vector $\mathbf{x} \in \mathbb{R}^d$ is first partitioned into $d/3$ sub-vectors, $\{\mathbf{v}_i\}_{i=1}^{d/3}$, where each $\mathbf{v}_i = (v_x, v_y, v_z) \in \mathbb{R}^3$. Each sub-vector $\mathbf{v}_i$ is then "lifted" into the quaternion space $\mathbb{H}$ as a pure quaternion (i.e., a quaternion with a zero scalar part):

$$\mathbf{p} = 0 + v_x\mathbf{i} + v_y\mathbf{j} + v_z\mathbf{k} \tag{1}$$

Given a unit quaternion $\mathbf{r}$ that represents a desired rotation, the transformation of $\mathbf{p}$ is given by the sandwich product:

$$\mathbf{p}' = \mathbf{rpr}^* \tag{2}$$

where $\mathbf{r}^*$ is the conjugate of $\mathbf{r}$, which for a unit quaternion is equivalent to its inverse ($\mathbf{r}^{-1}$). A crucial property of this operation is that the result $\mathbf{p}'$ remains a pure quaternion. Its vector part corresponds to the rotated vector $\mathbf{v}'_i$ in $\mathbb{R}^3$. This rotational operation is, by construction, an isometry for each 3D sub-vector, preserving its norm $\|\mathbf{v}_i\|$.

The rotational quaternion $\mathbf{r}$ is a function of positional indices, e.g., $(h, w)$ for a 2D image, which encode phase information $\theta_h$ and $\theta_w$. For a position $(p_h, p_w)$ and a given sub-vector $i \in \{1, \ldots, d/3\}$,

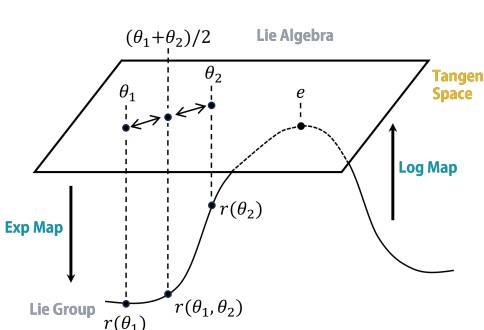
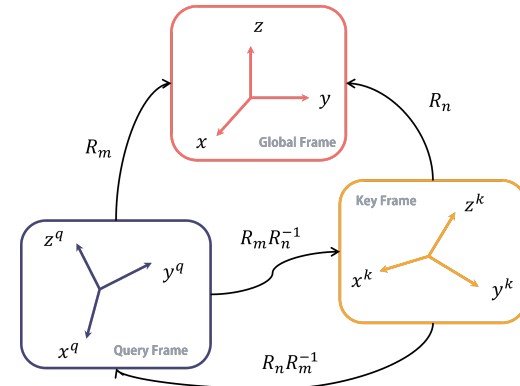

(a) **The Log-Exp average of Lie Algebra and Lie Group.** To ensure symmetry, non-commutative rotations $r(\theta_1), r(\theta_2)$ are mapped to the linear Lie Algebra (Tangent Space, where $e$ is identity element) via the Log Map. An arithmetic mean $(\theta_1 + \theta_2)/2$ is computed, and the result is mapped back to the Lie Group using the Exp Map to produce the symmetric operator $r(\theta_1, \theta_2)$.

(b) **The transform of Global Frame and Relative frame.** This panel presents two interpretations of the attention score $\langle R_m q, R_n k \rangle$. Global Frame: $R_m$ and $R_n$ transform vectors $(q, k)$ into a shared, absolute Global Frame. Relative Frame: $R_m R_n^{-1}$ is the relative rotation operator that transforms the Key Frame (at $n$) into the Query Frame (at $m$).

Figure 2: Illustration of mathematical structure and coordinate transform.

these are defined as $\theta_h = p_h \cdot \lambda^{2i/d}$ and $\theta_w = p_w \cdot \lambda^{2i/d}$, where $\lambda$ is a chosen base which is set as $\lambda = 100$ as usual (Heo et al., 2024).

## 3.2 Constructing a Symmetric Operator

For 2D data, positional information along the height and width dimensions can be encoded as rotations about distinct axes. A natural choice is to associate them with rotations about the y-axis ($\mathbf{j}$) and z-axis ($\mathbf{k}$), respectively. This yields two base quaternions:

$$\mathbf{r}_h(\theta_h) = \cos\left(\frac{\theta_h}{2}\right) + \sin\left(\frac{\theta_h}{2}\right)\mathbf{j}, \quad \mathbf{r}_w(\theta_w) = \cos\left(\frac{\theta_w}{2}\right) + \sin\left(\frac{\theta_w}{2}\right)\mathbf{k}$$

A naive composition of these rotations via quaternion multiplication, such as $\mathbf{r}_{hw} = \mathbf{r}_h \mathbf{r}_w$, is ill-suited for our purpose. Quaternion multiplication is non-commutative ($\mathbf{r}_h \mathbf{r}_w \neq \mathbf{r}_w \mathbf{r}_h$), meaning the resulting rotation would be arbitrarily dependent on the chosen order of operations, creating an undesirable symmetric bias between the height and width encodings. (This requirement for symmetry is crucial because GeoPE is specifically designed for structures like $2D$ images, where the spatial axes are fundamentally isotropic and no axis is privileged, thus necessitating a commutative operator for consistent geometric coupling.)

To construct an operator that treats each spatial dimension symmetrically, we turn to the tools of Lie theory. The core idea is to compute the geometric mean of the rotations. This is achieved by mapping the quaternions from the non-linear Lie group of 3D rotations, SO(3), to its corresponding linear Lie algebra, $\mathfrak{so}(3)$, via the logarithm map. In this tangent vector space, a simple averaging operation is well-defined and commutative. The result is then mapped back to the Lie group via the exponential map.

Accordingly, we define our symmetric rotational operator $\mathbf{r}$ as:

$$\mathbf{r}(\theta_h, \theta_w) = \exp\left(\frac{1}{2}\left(\log(\mathbf{r}_h(\theta_h)) + \log(\mathbf{r}_w(\theta_w))\right)\right) \tag{3}$$

This symmetric coupling ensures that the relative position is not merely a linear combination of independent axes, but a unified geometric transformation. This prevents the model from collapsing the 2D structure back into 1D sequence patterns. As derived in Appendix C, the intermediate averaged vector in the Lie algebra $\mathfrak{so}(3)$ is $(0, \theta_h/4, \theta_w/4)$. The exponential map yields an elegant

closed-form solution for the resulting quaternion:

$$\mathbf{r} = \cos\left(\frac{\Theta}{2}\right) + \sin\left(\frac{\Theta}{2}\right)\frac{\theta_h}{2\Theta}\mathbf{j} + \sin\left(\frac{\Theta}{2}\right)\frac{\theta_w}{2\Theta}\mathbf{k} \tag{4}$$

where $\Theta = \frac{1}{2}\sqrt{\theta_h^2 + \theta_w^2}$. The coupled phase $\Theta$ is proportional to the Euclidean distance between $(\theta_h, \theta_w)$ and the origin, while the vector components ensure that the influence of each positional phase remains monotonic. As illustrated in Figure 2a, this log-exp average provides a commutative and geometrically sound method for combining rotations.

The quaternion rotation in Equation 2 is equivalent to a matrix-vector product, $\mathbf{v}' = \mathbf{R}\mathbf{v}$, where $\mathbf{R} \in$ SO(3) is the rotation matrix corresponding to $\mathbf{r}$. The complete transformation on a $d$-dimensional query vector $\mathbf{q}$ or key vector $\mathbf{k}$ is thus a block-diagonal matrix:

$$\mathbf{R}_{\text{GeoPE}} = \begin{pmatrix} \mathbf{R}_1 & \mathbf{0} & \cdots & \mathbf{0} \\ \mathbf{0} & \mathbf{R}_2 & \cdots & \mathbf{0} \\ \vdots & \vdots & \ddots & \vdots \\ \mathbf{0} & \mathbf{0} & \cdots & \mathbf{R}_{d/3} \end{pmatrix}, \mathbf{R}_i = \begin{pmatrix} \cos(\Theta) & -\frac{\theta_w \sin(\Theta)}{\sqrt{\theta_h^2+\theta_w^2}} & \frac{\theta_h \sin(\Theta)}{\sqrt{\theta_h^2+\theta_w^2}} \\ \frac{\theta_w \sin(\Theta)}{\sqrt{\theta_h^2+\theta_w^2}} & 1-\frac{\theta_w^2(1-\cos(\Theta))}{\theta_h^2+\theta_w^2} & \frac{\theta_h\theta_w(1-\cos(\Theta))}{\theta_h^2+\theta_w^2} \\ -\frac{\theta_h \sin(\Theta)}{\sqrt{\theta_h^2+\theta_w^2}} & \frac{\theta_h\theta_w(1-\cos(\Theta))}{\theta_h^2+\theta_w^2} & 1-\frac{\theta_h^2(1-\cos(\Theta))}{\theta_h^2+\theta_w^2} \end{pmatrix}$$

where each $\mathbf{R}_i$ is a $3 \times 3$ rotation matrix derived from the quaternion $\mathbf{r}$ computed with phases $(\theta_{h,i}, \theta_{w,i})$ specific to that block. When the structured tensor is one-dimensional, GeoPE as discussed in Appendix F gracefully degenerates to a 2D rotation equivalent to the original RoPE (Su et al., 2024). Meanwhile, GeoPE keep long distance decay with projected similarity in Equation9 as disscussed in Appendix E

### 3.3 EXTENSION TO THREE SPATIAL DIMENSIONS

The GeoPE framework extends naturally to three spatial dimensions (e.g., for video data or volumetric scans) with positions $(d, h, w)$. We introduce a third base quaternion for depth, $\mathbf{r}_d(\theta_d) = \cos(\frac{\theta_d}{2}) + \sin(\frac{\theta_d}{2})\mathbf{i}$, and compute the symmetric average of the three rotations:

$$\mathbf{r}(\theta_d, \theta_h, \theta_w) = \exp\left(\frac{1}{3}\left(\log(\mathbf{r}_d) + \log(\mathbf{r}_h) + \log(\mathbf{r}_w)\right)\right) \tag{5}$$

This yields the three-dimensional GeoPE operator by results in Appendix G:

$$\mathbf{r} = \cos\left(\frac{\Theta}{2}\right) + \sin\left(\frac{\Theta}{2}\right)\left(\frac{\theta_d}{3\Theta}\mathbf{i} + \frac{\theta_h}{3\Theta}\mathbf{j} + \frac{\theta_w}{3\Theta}\mathbf{k}\right) \tag{6}$$

where the composite phase is now $\Theta = \frac{1}{3}\sqrt{\theta_d^2 + \theta_h^2 + \theta_w^2}$. This demonstrates the flexibility and scalability of our proposed geometric approach.

### 3.4 LINEAR FORMULATION FOR RELATIVE POSITION ENCODING

A critical property of positional embeddings in Transformer architectures is the ability to encode relative position, as the attention mechanism is fundamentally relational. For a query $\mathbf{q}$ at position $m$ and a key $\mathbf{k}$ at position $n$, the attention score is a function of $\langle \mathbf{R}_m \mathbf{q}, \mathbf{R}_n \mathbf{k} \rangle = \langle \mathbf{q}, \mathbf{R}_m^\top \mathbf{R}_n \mathbf{k} \rangle$. Ideally, the relative rotation matrix $\mathbf{R}_{m \to n} = \mathbf{R}_m^\top \mathbf{R}_n$ should depend only on the displacement $n - m$.

Our symmetric operator, while geometrically sound, does not inherently guarantee this linear relationship in the parameter space. That is, $\mathbf{r}(\theta_h, \theta_w) \neq \mathbf{r}(\phi_h, \phi_w)\mathbf{r}(\theta_h - \phi_h, \theta_w - \phi_w)$. To recover an inductive bias analogous to the simple phase subtraction in 1D RoPE, we propose a 'Linear GeoPE' formulation. The core insight is to enforce a linear relationship in the Lie algebra, where rotational composition is approximated by vector addition. By defining the relative rotation based on the difference of the Lie algebra vectors, i.e., $\mathbf{u}_{\text{rel}} = \mathbf{u}_k - \mathbf{u}_q$, we ensure the resulting rotation depends on the simple linear displacement of positional phases, mirroring the behavior of the original RoPE.

Let the Lie algebra vectors for a query at position $(h_q, w_q)$ and a key at position $(h_k, w_k)$ be $\mathbf{u}_q = (0, \theta_{h_q}/4, \theta_{w_q}/4)$ and $\mathbf{u}_k = (0, \theta_{h_k}/4, \theta_{w_k}/4)$, respectively. We define the relative Lie algebra vector as their difference:

$$\mathbf{u}_{\text{rel}} = \mathbf{u}_k - \mathbf{u}_q = \left(0, \frac{\theta_{h_k} - \theta_{h_q}}{4}, \frac{\theta_{w_k} - \theta_{w_q}}{4}\right) \tag{7}$$

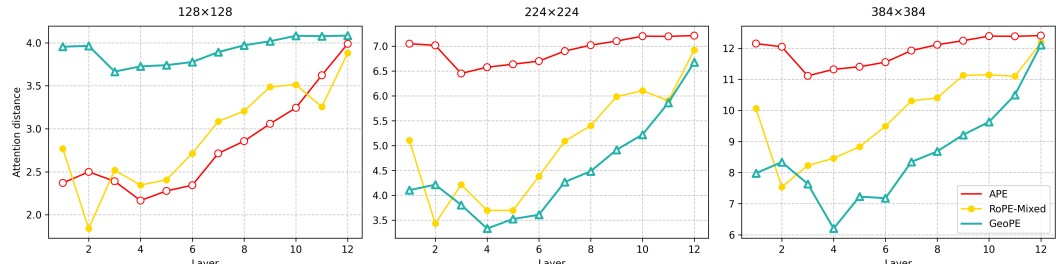

Figure 3: **Mean attention distance as a function of layer depth across different input resolutions.** The distance is computed as the average over attention scores, where query–key spatial distances are weighted by their corresponding attention weights and then normalized. While all methods exhibit an expanding receptive field in deeper layers, APE's consistently higher distance suggests an inefficient and unfocused global search. In contrast, GeoPE maintains a more moderate distance, indicating a more structured and efficient strategy for balancing local and global information gathering. These relative trends remain consistent across all tested resolutions.

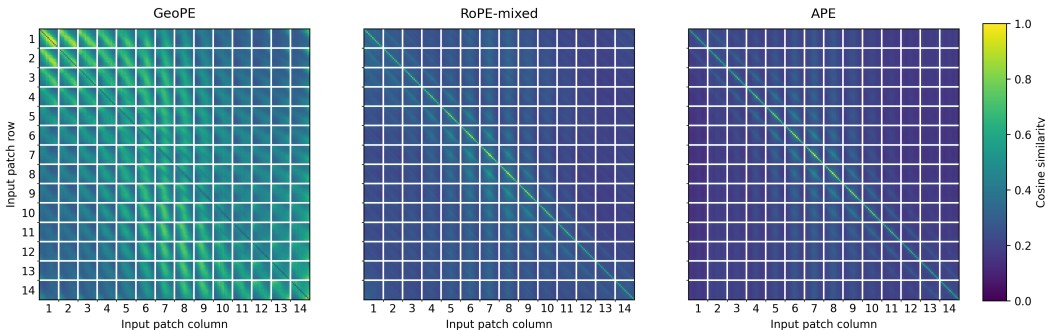

Figure 4: **Attention Map Visualization.** This figure compares the self-attention patterns from the final layer of ViT-Base models, evaluated after pre-training from scratch on ImageNet-1K. The heatmaps visualize the cosine similarity between patch representations, averaged across all attention heads, where the fine-grained patterns within the large squares reflect the feature correlation and similarity among the pixels inside each input patch. APE results in highly localized attention focused on the diagonal. RoPE-mixed shows a more distributed local pattern. In contrast, GeoPE facilitates complex, long-range attention, indicating a significantly more global receptive field. GeoPE's global attention pattern demonstrates its improved ability to integrate features across the entire image based on geometric structure.

The relative rotation is then obtained by mapping this difference back to the Lie group: $\mathbf{r}_{\text{rel}} = \exp(\mathbf{u}_{\text{rel}})$. This construction ensures that the transformation between any two positions depends solely on their relative displacement.

This allows the attention score to be computed as $\langle \mathbf{q}, \mathbf{R}_{\text{rel}}\mathbf{k} \rangle$. However, unlike the 1D case where the relative rotation matrix is a simple 2D rotation, the $3 \times 3$ matrix $\mathbf{R}_{\text{rel}}$ is generally dense. Applying this transformation explicitly is computationally more demanding than the standard GeoPE formulation, presenting a trade-off between enforcing a strict linear inductive bias and computational efficiency.

## 4 DISCUSSION

In this section, we further explore the properties of GeoPE to provide a deeper understanding of its mechanism and impact. We analyze the geometric interpretation of the attention score under 3D rotations and discuss how GeoPE influences the model's spatial reasoning capabilities.

## 4.1 Geometric Interpretation of the GeoPE

GeoPE enriches the self-attention mechanism by incorporating a geometrically meaningful understanding of space. The attention score between a query $\mathbf{q}$ at position $m = (h_m, w_m)$ and a key $\mathbf{k}$ at position $n = (h_n, w_n)$ is computed on their rotated counterparts:

$$\text{AttnScore}(\mathbf{q}_m, \mathbf{k}_n) = \langle \mathbf{R}_m \mathbf{q}, \mathbf{R}_n \mathbf{k} \rangle = \langle \mathbf{q}, \mathbf{R}_m^\top \mathbf{R}_n \mathbf{k} \rangle \tag{8}$$

This formulation offers two powerful, complementary geometric interpretations as shown in Figure 2b.

**Global Coordinate Frame.** One perspective is that $\mathbf{R}_m$ and $\mathbf{R}_n$ transform the query and key vectors from their local, position-agnostic feature spaces into a shared global coordinate frame defined by their absolute positions. The inner product is then computed in this global frame, allowing for a direct, spatially-aware comparison.

**Relative Coordinate Frame.** Alternatively, and perhaps more intuitively for attention, the term $\mathbf{R}_{\text{rel}} = \mathbf{R}_m^\top \mathbf{R}_n$ can be interpreted as a relative rotation operator. It transforms the key vector $\mathbf{k}$ from its own positional frame at $n$ into the query's positional frame at $m$. The attention score is thus a measure of feature similarity after aligning the key to the query's geometric context.

Unlike the simple phase difference in Heo et al. (2024), this 3D relative rotation depends not only on the magnitude of the displacement $(h_n - h_m, w_n - w_m)$ but also on the direction of displacement. The attention score is governed by the inner product of a vector with its rotated version, which, according to Rodrigues' rotation formula, is a function of both the angle and the axis of this relative rotation. For a rotation of angle $A$ about an axis $\mathbf{n}$, the inner product as discussed in Appendix D becomes:

$$\langle \mathbf{q}, \mathbf{R}_{\text{rel}} \mathbf{k} \rangle = \underbrace{\langle \mathbf{q}, \mathbf{k} \rangle \cos(A)}_{\text{Projected Similarity}} + \underbrace{(\mathbf{q} \cdot \mathbf{n})(\mathbf{k} \cdot \mathbf{n})(1 - \cos(A))}_{\text{Axial Alignment}} - \underbrace{(\mathbf{n} \times \mathbf{q}) \cdot \mathbf{k} \sin(A)}_{\text{Torsional Component}} \tag{9}$$

This decomposition provides a clear geometric intuition. The Projected Similarity term generalizes RoPE by modulating similarity based on displacement magnitude (via angle A). Crucially, the Axial Alignment term adds sensitivity to the direction of displacement (via axis $n$). Unlike 1D-based methods that primarily encode scalar distance (which can be misleading due to flattening), this term allows the attention mechanism to explicitly differentiate between vertical and horizontal relationships.

Consequently, the Torsional Component captures the relative spatial orientation. This equips the model with a geometric directional prior, enabling it to recognize shape boundaries defined by specific directional transitions (e.g., corners and edges) rather than just local texture continuity found in the flattened sequence.

For Linear GeoPE, the angle $A_i$ and axis $\mathbf{n}_i$ for the i-th sub-vector are defined as:

$$A_i = \frac{1}{2}\sqrt{(\Delta\theta_{h,i})^2 + (\Delta\theta_{w,i})^2}, \ \mathbf{n}_i = \frac{\frac{\Delta\theta_{h,i}}{4}\mathbf{j} + \frac{\Delta\theta_{w,i}}{4}\mathbf{k}}{\frac{1}{4}\sqrt{(\Delta\theta_{h,i})^2 + (\Delta\theta_{w,i})^2}} = \frac{\Delta\theta_{h,i}\mathbf{j} + \Delta\theta_{w,i}\mathbf{k}}{\sqrt{(\Delta\theta_{h,i})^2 + (\Delta\theta_{w,i})^2}}$$

where $\Delta\theta_{h,i} = \theta_{h_k,i} - \theta_{h_q,i} = (p_{h_k} - p_{h_q}) \cdot \lambda^{2i/d} = \Delta p_h \cdot \lambda^{2i/d}$ and $\Delta\theta_{w,i} = \theta_{w_k,i} - \theta_{w_q,i} = (p_{w_k} - p_{w_q}) \cdot \lambda^{2i/d} = \Delta p_w \cdot \lambda^{2i/d}$. This shows that the interaction is a complex blend of the original similarity $\langle \mathbf{q}, \mathbf{k} \rangle$ and terms modulated by the alignment of the vectors with the relative rotation axis, endowing the model with a richer, more expressive spatial bias.

## 4.2 Impact on Attention Patterns and Spatial Awareness

We hypothesize that GeoPE's geometric inductive bias fosters more effective spatial reasoning by enabling more meaningful attention patterns. Our analyses support this: models equipped with GeoPE exhibit longer attention distances in Figure 3 and more global attention maps in Figure 4. This behavior allows the model to capture long-range dependencies and integrate information across the entire spatial domain, rather than focusing only on local texture. We posit that this enhanced global awareness directly contributes to the performance gains and improved shape-texture bias observed in our experiments.

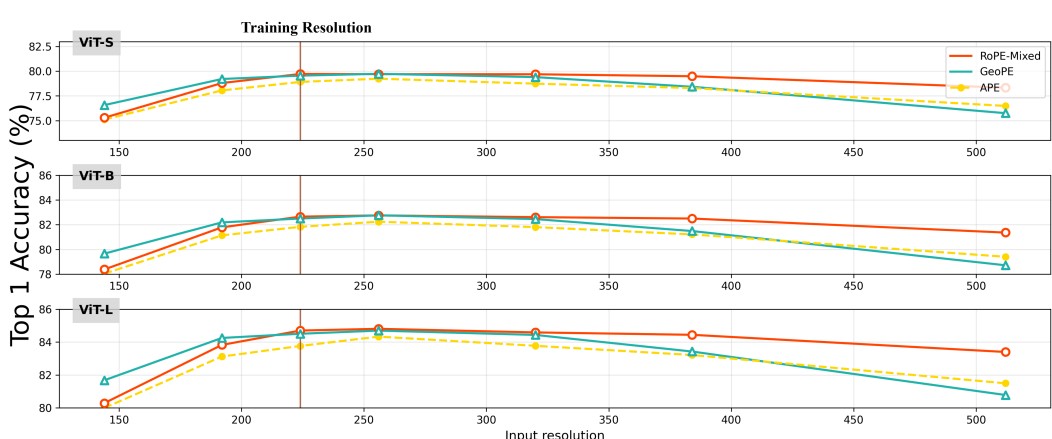

Figure 5: **Generalization performance to unseen input resolutions for ViT-S, -B, and -L models.** All models are trained at a fixed 224x224 resolution (marked by the vertical line) and evaluated on a range of different resolutions. Absolute Positional Embedding (APE) fails to generalize, with its accuracy collapsing at higher resolutions. In contrast, relative embeddings like RoPE-Mixed and GeoPE show strong robustness as their performance degrades gracefully, highlighting their suitability for real-world applications with variable input sizes.

## 5 EXPERIMENTS

We validate our methods, GeoPE and Linear GeoPE, through comprehensive experiments on image classification, object detection, and 3D semantic segmentation, benchmarking them against standard baselines and existing 2D rotational embeddings.

### 5.1 IMAGE CLASSIFICATION

We evaluate our methods on the ImageNet-1K classification task using Vision Transformer (ViT) (Dosovitskiy et al., 2020) and Swin Transformer (Liu et al., 2021) backbones, following the DeiT3 training protocol (Armeni et al., 2016b) with CE loss with fixed random seed(3407) Picard (2023). We additionally provide more reliable experiments in the Appendix K.

As shown in Table 1, GeoPE consistently improves Top-1 accuracy across all backbones. It outperforms standard baselines like APE and CPE (Chu et al., 2021) on ViT models and matches or exceeds the performance of PRB and Rope-Mixed (Heo et al., 2024) on Swin Transformers, demonstrating the broad applicability of its geometric prior. Furthermore, as depicted in Figure 5, Linear GeoPE exhibits exceptional zero-shot inference capabilities across multiple resolutions, confirming its superior extrapolation properties as a natural high-dimensional extension of RoPE (Su et al., 2024).

### 5.2 OBJECT DETECTION

To assess GeoPE's impact on tasks requiring fine-grained spatial awareness, we evaluate it on the MS-COCO (Lin et al., 2014) object detection benchmark. We integrate GeoPE into the DINO-ViTDet (Zhang et al., 2022) framework, a strong object detection pipeline.

Table 2 shows that GeoPE consistently improves mAP for both ViT-B and ViT-L backbones. Compared with APE and Rope-Mixed (Heo et al., 2024), GeoPE provides the largest relative gains, highlighting the importance of explicit geometric priors in capturing global spatial relationships critical for accurate object detection.

Table 1: Comparison of different Positional Encodings (PE) on ImageNet-1K. ViTs follow the recipe protocol, and Swin Transformers follow their original protocol. **Bold** denotes the best result in each group.

| Backbone | Resolution | PE Method | Top-1 Acc |
|----------|-----------|-----------|-----------|
| ViT-Small | $192 \times 192$ | GeoPE | 78.5 |
|  | $192 \times 192$ | LinGeoPE | 78.8 |
|  | $224 \times 224$ | APE | 79.9 |
|  | $224 \times 224$ | CPE | 80.7 |
|  | $224 \times 224$ | GeoPE | **81.2** |
| ViT-Base | $224 \times 224$ | APE | 81.3 |
|  | $224 \times 224$ | CPE | 82.2 |
|  | $224 \times 224$ | GeoPE | **82.5** |
| ViT-Large | $224 \times 224$ | APE | 83.3 |
|  | $224 \times 224$ | CPE | 83.6 |
|  | $224 \times 224$ | GeoPE | **83.9** |
| Swin-S | $224 \times 224$ | RPB | 83.0 |
|  | $224 \times 224$ | Rope-Mixed | 83.4 |
|  | $224 \times 224$ | GeoPE | **83.5** |
| Swin-B | $224 \times 224$ | RPB | 83.5 |
|  | $224 \times 224$ | Rope-Mixed | **83.8** |
|  | $224 \times 224$ | GeoPE | 83.6 |

## 5.3 3D SEMANTIC SEGMENTATION

To verify the hypothesis that GeoPE is suitable for any structured tensor data where spatial relationships are paramount, we apply it to 3D point cloud segmentation on the S3DIS dataset (Armeni et al., 2016b). We incorporate GeoPE into the Point Transformer architecture.

As reported in Table 3, GeoPE improves all major metrics, including overall accuracy, mean class accuracy, and mean IoU, relative to the RPE baseline. These improvements confirm that explicitly encoding multi-axis spatial relationships allows the model to better capture 3D geometric structures, validating the general applicability of GeoPE beyond 2D vision tasks.

## 5.4 SHAPE-TEXTURE BIAS ANALYSIS

To provide a deeper insight into how GeoPE enhances spatial reasoning, we conduct an analysis of the model's shape-texture bias. A strong shape bias—the tendency to prioritize global object structure over local texture in decision-making—is a critical characteristic correlated with superior robustness and generalization capabilities. We assess this property using the rigorous methodology proposed by Geirhos et al. (2018) , which employs specially constructed cue-conflict stimuli (images where texture and object shape point to conflicting categories) to explicitly quantify the model's decision preference.

As illustrated in Figure 6, GeoPE consistently shifts the model towards a stronger Shape Bias. Standard positional encodings often overfit to texture because the flattened sequence preserves local texture statistics even across unnatural boundaries (like row edges).

By enforcing a strict 3D geometric coupling, GeoPE penalizes attention to these 'false sequence neighbors' and rewards alignment with the true 2D structure. This result confirms that our method successfully mitigates the topological disruption of flattening, transitioning the model from a texture-biased sequence learner to a shape-aware geometric learner.

Table 2: This table reports MSCOCO(Lin et al., 2014) detection performance (box AP). DINO(Zhang et al., 2022) is trained under the 12-epoch DINO-ViTDet setting(Ren et al., 2023). For GeoPE, we apply it to the ViT backbone, which is pre-trained on ImageNet-1K using the 400-epoch DeiT-III recipe.

| Backbone | PE | mAP |
|---|---|---|
| | APE | 49.4 |
| ViT-base | Rope-Mixed | 51.2 |
| | GeoPE | **51.3** |
| | APE | 51.1 |
| ViT-large | Rope-Mixed | 52.9 |
| | GeoPE | **53.1** |

Table 3: Semantic segmentation performance on the S3DIS dataset(Armeni et al., 2016a), evaluated using 6-fold cross-validation.

| Backbone | PE | OA | mAcc | mIoU |
|---|---|---|---|---|
| Point-Transformer | RPE | 90.2 | 81.9 | 73.5 |
| | GeoPE | **90.5** | **82.1** | **74.4** |

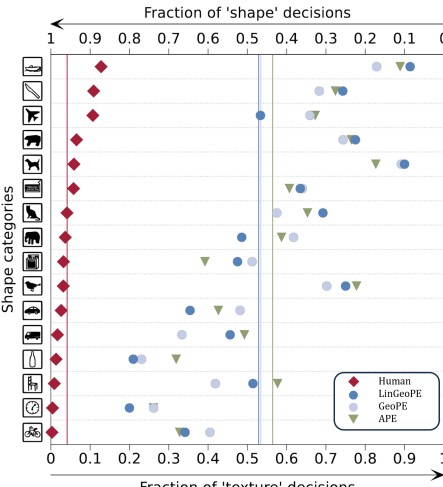

Figure 6: **Shape Bias Relation Analysis.** This figure analyzes the decision bias of ViT-Small models, pre-trained from scratch on ImageNet-1K, using a cue-conflict methodology. The plot compares the fraction of 'shape' decisions (Y-axis) against 'texture' decisions (X-axis) for stimuli where these visual cues conflict. GeoPE and LinGeoPE consistently shift the model's bias towards shape , aligning them closer to human perception and suggesting a more robust, holistic visual understanding.

## 6    CONCLUSION

We propose GeoPE, a framework designed to restore the natural spatial topology disrupted by the flattening operation in Vision Transformers. By lifting coordinates into 3D Euclidean space using quaternions, GeoPE introduces a geometrically coupled encoding that effectively distinguishes true spatial locality from false sequence adjacency. To handle the non-commutativity of quaternions, we develop a symmetric averaging technique based on Lie theory and derive a Linear GeoPE variant that preserves relative position inductive biases. Extensive experiments demonstrate that GeoPE not only boosts performance on 2D and 3D tasks but also significantly enhances shape bias, confirming that the model has transitioned from relying on local texture statistics to understanding global geometry. Our work offers a principled path for robust spatial modeling in structured tensor data.

## REPRODUCIBILITY STATEMENT

We have made extensive efforts to ensure the reproducibility of our work. The main paper provides detailed descriptions of the proposed method, model architectures, and training procedures. Additional experimental details, ablation studies, and theoretical derivations are included in the Appendix. We also provide the complete data preprocessing steps and hyperparameter configurations in the supplementary material. Furthermore, we submit the anonymized source code and training scripts as supplementary material to facilitate replication of all reported results.

## ETHICAL STATEMENT

We used a large language model to improve the grammar and clarity of the paper's text. All research ideas, experiments, and analyses are our own.

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

## A  QUATERNION ROTATIONAL TRANSFORMATIONS

Quaternions are four-dimensional hypercomplex numbers that can be used to represent rotations in three-dimensional space. A quaternion contains three imaginary components.

Its standard form is:

$$\mathbf{p} = w + x\mathbf{i} + y\mathbf{j} + z\mathbf{k} \tag{10}$$

It can also be written more compactly as:

$$\mathbf{p} = s + \mathbf{v} \tag{11}$$

The basic properties of quaternions are:

$$\mathbf{i}^2 = \mathbf{j}^2 = \mathbf{k}^2 = -1 \tag{12}$$
$$\mathbf{ij} = -\mathbf{ji} = \mathbf{k} \tag{13}$$
$$\mathbf{jk} = -\mathbf{kj} = \mathbf{i} \tag{14}$$
$$\mathbf{ki} = -\mathbf{ik} = \mathbf{j} \tag{15}$$

We can see that, unlike real or complex numbers, quaternions satisfy anticommutative relations rather than commutative ones, and therefore their multiplication is non-commutative.For example, for two quaternions $\mathbf{p}_1$ and $\mathbf{p}_2$, we have

$$\mathbf{p}_1\mathbf{p}_2 \neq \mathbf{p}_2\mathbf{p}_1. \tag{16}$$

Let two quaternions be: $\mathbf{p}_1 = s_1 + \mathbf{v}_1 \quad \mathbf{p}_2 = s_2 + \mathbf{v}_2$, their multiplication formula is:

$$\mathbf{p}_1 \mathbf{p}_2 = s_1 s_2 - \mathbf{v}_1 \cdot \mathbf{v}_2 + s_1 \mathbf{v}_2 + s_2 \mathbf{v}_1 + \mathbf{v}_1 \times \mathbf{v}_2 \tag{17}$$

where $\cdot$ denotes the dot product and $\times$ denotes the cross product.

A rotation in three-dimensional space can be regarded as a function $\phi$, which is a mapping from $\mathbb{R}^3$ to itself. For the function $\phi$ to represent a rotation, it must preserve vector lengths, angles, and handedness during the transformation.

To preserve lengths, it must satisfy:

Length is preserved.

$$\|\phi(\mathbf{p})\| = \|\mathbf{p}\| \tag{18}$$

Angles is preserved.

$$\phi(\mathbf{p}_1) \cdot \phi(\mathbf{p}_2) = \mathbf{p}_1 \cdot \mathbf{p}_2 \tag{19}$$

Handedness is preserved.

$$\phi(\mathbf{p}_1) \times \phi(\mathbf{p}_2) = \phi(\mathbf{p}_1 \times \mathbf{p}_2) \tag{20}$$

Through formula derivation and verification, the following function is shown to satisfy the above conditions for representing quaternion rotation.

$$\phi_{\mathbf{r}}(\mathbf{p}) = \mathbf{r}\mathbf{p}\mathbf{r}^{-1} \tag{21}$$

Here, $\mathbf{r}$ is a non-zero quaternion, and the argument $\mathbf{p}$ of the function can be viewed as a point in three-dimensional space, that is, a quaternion with a real (or scalar) part equal to zero.

Next, we need to find the expression for the quaternion $\mathbf{r}$ such that it corresponds to a rotation transformation around the rotation axis $\mathbf{A}$ by an angle $\theta$. After derivation, a unit quaternion $\mathbf{r}$ can be chosen, and the expression for $\mathbf{r}$ is:

$$\mathbf{r} = \cos\frac{\theta}{2} + \sin\frac{\theta}{2}\mathbf{A} \tag{22}$$

where A is usually represented by i, j, and k.

In summary, to apply a rotation transformation to a three-dimensional point $\mathbf{p}$, which is treated as a quaternion with a real part of zero, also known as a pure quaternion and imaginary quaternion, via the quaternion $\mathbf{r}$, one only needs to perform the following calculation:

$$\mathbf{p}' = \mathbf{r}\mathbf{p}\mathbf{r}^{-1} \tag{23}$$

We note that for any non-zero scalar $a$ (e.g., $a = -1$), the quaternions $a\mathbf{r}$ and $\mathbf{r}$ represent the same rotation. This is proven as follows:

$$(a\mathbf{r})\,\mathbf{p}\,(a\mathbf{r})^{-1} = a\mathbf{r}\,\mathbf{p}\,\frac{\mathbf{r}^{-1}}{a} = \mathbf{r}\,\mathbf{p}\,\mathbf{r}^{-1}. \tag{24}$$

Furthermore, the product of two quaternions, $\mathbf{r}_1$ and $\mathbf{r}_2$, also represents a rotation. Specifically, $\mathbf{r}_1\mathbf{r}_2$ represents the rotation obtained by first applying the rotation $\mathbf{r}_2$, followed by $\mathbf{r}_1$. The proof is given by:

$$\mathbf{r}_1(\mathbf{r}_2\,\mathbf{p}\,\mathbf{r}_2^{-1})\mathbf{r}_1^{-1} = (\mathbf{r}_1\mathbf{r}_2)\,\mathbf{p}\,(\mathbf{r}_2^{-1}\mathbf{r}_1^{-1}) = (\mathbf{r}_1\mathbf{r}_2)\,\mathbf{p}\,(\mathbf{r}_1\mathbf{r}_2)^{-1}. \tag{25}$$

This property allows us to concatenate an arbitrary number of rotation quaternions into a single quaternion.

From the above, we can see that quaternion multiplication is not commutative. Thus, for two unit quaternions $\mathbf{r}_1$ and $\mathbf{r}_2$, we have

$$\mathbf{r}_1\mathbf{r}_2 \neq \mathbf{r}_2\mathbf{r}_1. \tag{26}$$

This problem can be addressed using Lie groups and Lie algebras.

## B    LIE GROUPS AND LIE ALGEBRAS

Lie groups are mathematical objects that possess both group structures and smooth manifold structures. Elements of a Lie group can undergo transformations in a continuous manner. Common examples of Lie groups include rotation groups $SO(n)$, special linear groups $SL(n, \mathbb{R})$, and general linear groups $GL(n, \mathbb{R})$. Among them, rotation groups $SO(n)$ describes rotational operations in an $n$-dimensional space. In particular, $SO(3)$ describes rotations in three-dimensional space.

Lie algebras are the tangent spaces of Lie groups, characterizing the local properties of Lie groups near the identity element. Each Lie group corresponds to a Lie algebra. Lie algebra elements generate Lie group elements through the exponential map. Conversely, Lie group elements can be mapped back to the Lie algebra through the logarithm map.

When the Lie group is a matrix group, elements of the Lie algebra typically correspond to infinitesimal variations of matrices. For example, the Lie algebra $\mathfrak{so}(3)$ of the rotation group $SO(3)$ can be represented using skew-symmetric matrices, which describe infinitesimal rotations in three-dimensional space.

The multiplication structure of quaternions has an analogous relationship with elements of the Lie algebra $\mathfrak{so}(3)$. By mapping quaternions to elements of $\mathfrak{so}(3)$ via the logarithm map, quaternion rotations can be described and computed using Lie algebra operations and also addresses the non-commutativity of quaternion multiplication.

## C    DERIVATION OF THE SYMMETRIC OPERATOR

This section details the derivation of the closed-form solution for the symmetric rotational operator $\mathbf{r}(\theta_h, \theta_w)$ introduced in Section 3.2.

Our goal is to compute the geometric mean of two base rotations, $\mathbf{r}_h(\theta_h)$ and $\mathbf{r}_w(\theta_w)$, using the log-exp map formalism:

$$\mathbf{r}(\theta_h, \theta_w) = \exp\left(\frac{1}{2}\left(\log(\mathbf{r}_h(\theta_h)) + \log(\mathbf{r}_w(\theta_w))\right)\right) \tag{27}$$

The logarithm map for a unit quaternion $\mathbf{r} = \cos(\alpha) + \sin(\alpha)\mathbf{n}$, where $\mathbf{n}$ is a unit vector, is given by $\log(\mathbf{r}) = \alpha\mathbf{n}$. The vector $\alpha\mathbf{n}$ is an element of the Lie algebra $\mathfrak{so}(3)$.

The base quaternions are:

$$\mathbf{r}_h(\theta_h) = \cos\left(\frac{\theta_h}{2}\right) + \sin\left(\frac{\theta_h}{2}\right)\mathbf{j} \tag{28}$$

$$\mathbf{r}_w(\theta_w) = \cos\left(\frac{\theta_w}{2}\right) + \sin\left(\frac{\theta_w}{2}\right)\mathbf{k} \tag{29}$$

Applying the logarithm map to each, we get:

$$\log(\mathbf{r}_h(\theta_h)) = \frac{\theta_h}{2}\mathbf{j} \tag{30}$$

$$\log(\mathbf{r}_w(\theta_w)) = \frac{\theta_w}{2}\mathbf{k} \tag{31}$$

In the vector space $\mathfrak{so}(3) \cong \mathbb{R}^3$, these correspond to the vectors $(0, \theta_h/2, 0)$ and $(0, 0, \theta_w/2)$.

We compute the arithmetic mean of these vectors in the Lie algebra:

$$\mathbf{u} = \frac{1}{2}\left(\log(\mathbf{r}_h) + \log(\mathbf{r}_w)\right) = \frac{1}{2}\left(\frac{\theta_h}{2}\mathbf{j} + \frac{\theta_w}{2}\mathbf{k}\right) = \frac{\theta_h}{4}\mathbf{j} + \frac{\theta_w}{4}\mathbf{k} \tag{32}$$

This corresponds to the vector $(0, \theta_h/4, \theta_w/4)$, as stated in the main text.

The exponential map for a Lie algebra vector $\mathbf{u}$ is given by $\exp(\mathbf{u}) = \cos(\|\mathbf{u}\|) + \sin(\|\mathbf{u}\|)\frac{\mathbf{u}}{\|\mathbf{u}\|}$.

First, we compute the norm of our averaged vector $\mathbf{u}$:

$$\|\mathbf{u}\| = \sqrt{\left(\frac{\theta_h}{4}\right)^2 + \left(\frac{\theta_w}{4}\right)^2} = \frac{1}{4}\sqrt{\theta_h^2 + \theta_w^2} \tag{33}$$

Let us define the coupled phase $\Theta = \frac{1}{2}\sqrt{\theta_h^2 + \theta_w^2}$. Then, $\|\mathbf{u}\| = \frac{\Theta}{2}$.

Next, we find the corresponding unit axis vector:

$$\frac{\mathbf{u}}{\|\mathbf{u}\|} = \frac{\frac{\theta_h}{4}\mathbf{j} + \frac{\theta_w}{4}\mathbf{k}}{\frac{1}{4}\sqrt{\theta_h^2 + \theta_w^2}} = \frac{\theta_h\mathbf{j} + \theta_w\mathbf{k}}{\sqrt{\theta_h^2 + \theta_w^2}} = \frac{\theta_h}{2\Theta}\mathbf{j} + \frac{\theta_w}{2\Theta}\mathbf{k} \tag{34}$$

Finally, applying the exponential map yields the desired symmetric operator:

$$\mathbf{r} = \exp(\mathbf{u}) = \cos\left(\frac{\Theta}{2}\right) + \sin\left(\frac{\Theta}{2}\right)\left(\frac{\theta_h}{2\Theta}\mathbf{j} + \frac{\theta_w}{2\Theta}\mathbf{k}\right) \tag{35}$$

This completes the derivation.

## D    INNER PRODUCT WITH ROTATED VECTORS

This section provides the derivation for the inner product of a vector $\mathbf{q}$ with a rotated vector $\mathbf{Rk}$, as presented in the discussion on the geometric interpretation of attention.

A rotation of a vector $\mathbf{k} \in \mathbb{R}^3$ by an angle $A$ around a unit axis vector $\mathbf{n} \in \mathbb{R}^3$ is given by Rodrigues' rotation formula:

$$\mathbf{Rk} = \mathbf{k}\cos(A) + (\mathbf{n} \times \mathbf{k})\sin(A) + \mathbf{n}(\mathbf{n} \cdot \mathbf{k})(1 - \cos(A)) \tag{36}$$

To find the attention score, we compute the inner product of a query vector $\mathbf{q}$ with this rotated key vector:

$$\langle \mathbf{q}, \mathbf{Rk} \rangle = \langle \mathbf{q}, \mathbf{k}\cos(A) + (\mathbf{n} \times \mathbf{k})\sin(A) + \mathbf{n}(\mathbf{n} \cdot \mathbf{k})(1 - \cos(A)) \rangle \tag{37}$$

By the linearity of the inner product, we can distribute $\mathbf{q}$ across the terms:

$$\langle \mathbf{q}, \mathbf{Rk} \rangle = \langle \mathbf{q}, \mathbf{k} \rangle \cos(A) + \langle \mathbf{q}, (\mathbf{n} \times \mathbf{k}) \rangle \sin(A) + \langle \mathbf{q}, \mathbf{n}(\mathbf{n} \cdot \mathbf{k}) \rangle (1 - \cos(A)) \tag{38}$$

The last term can be simplified:

$$\langle \mathbf{q}, \mathbf{n}(\mathbf{n} \cdot \mathbf{k}) \rangle = (\mathbf{q} \cdot \mathbf{n})(\mathbf{n} \cdot \mathbf{k}) \tag{39}$$

The middle term involves the scalar triple product, which satisfies the identity $\mathbf{a} \cdot (\mathbf{b} \times \mathbf{c}) = \mathbf{b} \cdot (\mathbf{c} \times \mathbf{a}) = \mathbf{c} \cdot (\mathbf{a} \times \mathbf{b})$. Let $\mathbf{a} = \mathbf{q}, \mathbf{b} = \mathbf{n}, \mathbf{c} = \mathbf{k}$. Then:

$$\langle \mathbf{q}, (\mathbf{n} \times \mathbf{k}) \rangle = \mathbf{k} \cdot (\mathbf{q} \times \mathbf{n}) = -\mathbf{k} \cdot (\mathbf{n} \times \mathbf{q}) \tag{40}$$

Substituting these back, we obtain the final expression for the inner product:

$$\langle \mathbf{q}, \mathbf{Rk} \rangle = \langle \mathbf{q}, \mathbf{k} \rangle \cos(A) + (\mathbf{q} \cdot \mathbf{n})(\mathbf{k} \cdot \mathbf{n})(1 - \cos(A)) - (\mathbf{n} \times \mathbf{q}) \cdot \mathbf{k}\sin(A) \tag{41}$$

Note: The sign of the final term may vary depending on the convention used for the scalar triple product permutation, but the geometric intuition remains the same. The version in the main text is a common variant.

## E    LONG-DISTANCE DECAY PROPERTY

A key inductive bias of RoPE, which GeoPE is designed to generalize, is the decay of attention scores over large relative distances. We demonstrate that GeoPE preserves this behavior by analyzing the structure of the attention score's dominant term. The leading term in the total attention score is the sum:

$$S = \sum_{i=0}^{d/3-1} \langle \mathbf{q}_i, \mathbf{k}_i \rangle \cos(A_i) \tag{42}$$

where the angle $A_i$ for the $i$-th sub-vector is proportional to the relative distance $||\Delta p||$ and a frequency term $\lambda^{2i/d}$:

$$A_i = \frac{1}{2}\sqrt{(\Delta p_h \cdot \lambda^{2i/d})^2 + (\Delta p_w \cdot \lambda^{2i/d})^2} = \frac{1}{2}||\Delta p||\lambda^{2i/d} \tag{43}$$

Let $D = \frac{1}{2}||\Delta p||$ be the effective distance, $c_i = \langle \mathbf{q}_i, \mathbf{k}_i \rangle$ be the feature similarity, and $\phi_i = \lambda^{2i/d}$. The sum is $S = \sum_{i=0}^{N-1} c_i \cos(D\phi_i)$, where $N = d/3$.

To show this sum decays with $D$, we apply summation by parts (Abel transformation). Let $h_i = c_i$ and $g_i = \cos(D\phi_i)$. Let $G_k = \sum_{i=0}^{k} g_i$ be the partial sum of the cosine terms. The total sum is:

$$S = \sum_{i=0}^{N-1} h_i g_i = h_{N-1}G_{N-1} - \sum_{i=0}^{N-2}(h_{i+1} - h_i)G_i \tag{44}$$

By the triangle inequality:

$$|S| \leq |h_{N-1}||G_{N-1}| + \sum_{i=0}^{N-2}|h_{i+1} - h_i||G_i| \tag{45}$$

Assuming the feature similarities $c_i$ are well-behaved (i.e., bounded and with small successive differences, a reasonable assumption for trained embeddings), the magnitude of $S$ is primarily controlled by the magnitude of the partial sums $|G_k|$.

The partial sum $G_k = \sum_{i=0}^{k} \cos(D\phi_i)$ is a sum of cosines with geometrically increasing frequencies ($\phi_i = \lambda^{2i/d}$). For a large distance $D$, the arguments $D\phi_i$ grow rapidly, causing the cosine terms to oscillate with increasing frequency. Such sums are bounded due to destructive interference. While a simple closed-form bound is not available as in the arithmetic case (original RoPE), the geometric progression of frequencies ensures that the terms do not align constructively, keeping $|G_k|$ bounded for any $k$. As $D \to \infty$, the oscillations become more rapid, strengthening the cancellation effect. This implies that the average magnitude of the attention score decays with distance, preserving the crucial inductive bias for locality, analogous to the property shown in (Su et al., 2024).

## F  DEGENERATION OF GEoPE TO 1D RoPE

For 1D sequential data, GeoPE gracefully degenerates to a formulation equivalent to the original RoPE. In a 1D setting, we only have a single position index, say $p$, and its corresponding phase is $\theta = p \cdot \lambda^{2i/d}$. Since there is only one spatial dimension, the log-exp averaging process is unnecessary. We can define a single base rotation directly. Following the original RoPE, this is a 2D rotation, which can be embedded in our 3D framework as a rotation around a single fixed axis (e.g., the y-axis, $\mathbf{j}$).

The rotational quaternion for a phase $\theta$ is simply:

$$\mathbf{r}(\theta) = \cos\left(\frac{\theta}{2}\right) + \sin\left(\frac{\theta}{2}\right)\mathbf{j} \tag{46}$$

In GeoPE, feature vectors are partitioned into 3D sub-vectors $\mathbf{v} = (v_x, v_y, v_z)$. For a 1D application, we can effectively work with 2D sub-vectors by setting one component to zero, e.g., $\mathbf{v} = (v_x, 0, v_z)$. This corresponds to a pure quaternion $\mathbf{p} = v_x \mathbf{i} + v_z \mathbf{k}$.

The rotation is applied via the sandwich product $\mathbf{p}' = \mathbf{r}(\theta)\mathbf{p}\mathbf{r}(\theta)^*$. The rotation matrix corresponding to $\mathbf{r}(\theta)$ is a pure rotation around the y-axis:

$$\mathbf{R}(\theta) = \begin{pmatrix} \cos(\theta) & 0 & \sin(\theta) \\ 0 & 1 & 0 \\ -\sin(\theta) & 0 & \cos(\theta) \end{pmatrix} \tag{47}$$

Applying this rotation to our 2D-like sub-vector $\mathbf{v}' = \mathbf{R}(\theta)\mathbf{v}$:

$$\begin{pmatrix} v'_x \\ 0 \\ v'_z \end{pmatrix} = \begin{pmatrix} \cos(\theta) & 0 & \sin(\theta) \\ 0 & 1 & 0 \\ -\sin(\theta) & 0 & \cos(\theta) \end{pmatrix} \begin{pmatrix} v_x \\ 0 \\ v_z \end{pmatrix} = \begin{pmatrix} v_x \cos(\theta) + v_z \sin(\theta) \\ 0 \\ -v_x \sin(\theta) + v_z \cos(\theta) \end{pmatrix} \tag{48}$$

This operation is a 2D rotation on the coordinates $(v_x, v_z)$. The original RoPE applies the matrix $\begin{pmatrix} \cos(\theta) & -\sin(\theta) \\ \sin(\theta) & \cos(\theta) \end{pmatrix}$ to a pair of features $(f_1, f_2)$. The resulting transformation is $\begin{pmatrix} v'_x \\ v'_z \end{pmatrix} = \begin{pmatrix} \cos(\theta) & \sin(\theta) \\ -\sin(\theta) & \cos(\theta) \end{pmatrix} \begin{pmatrix} v_x \\ v_z \end{pmatrix}$. By identifying $v_x$ with $f_1$ and $v_z$ with $f_2$, this is equivalent to the standard RoPE rotation matrix for an angle of $-\theta$. Thus, GeoPE contains RoPE as a special case, differing only by a sign convention on the rotation angle.

## G  THREE DIMENSION EXTENSION

This section details the derivation for the 3D symmetric rotational operator, extending the logic from Appendix C.

For 3D data with positions $(d, h, w)$, we have three base quaternions corresponding to rotations about the $\mathbf{i}$, $\mathbf{j}$, and $\mathbf{k}$ axes:

$$\mathbf{r}_d(\theta_d) = \cos\left(\frac{\theta_d}{2}\right) + \sin\left(\frac{\theta_d}{2}\right)\mathbf{i} \tag{49}$$

$$\mathbf{r}_h(\theta_h) = \cos\left(\frac{\theta_h}{2}\right) + \sin\left(\frac{\theta_h}{2}\right)\mathbf{j} \tag{50}$$

$$\mathbf{r}_w(\theta_w) = \cos\left(\frac{\theta_w}{2}\right) + \sin\left(\frac{\theta_w}{2}\right)\mathbf{k} \tag{51}$$

We map these to the Lie algebra $\mathfrak{so}(3)$:

$$\log(\mathbf{r}_d) = \frac{\theta_d}{2}\mathbf{i} \tag{52}$$

$$\log(\mathbf{r}_h) = \frac{\theta_h}{2}\mathbf{j} \tag{53}$$

$$\log(\mathbf{r}_w) = \frac{\theta_w}{2}\mathbf{k} \tag{54}$$

We compute the arithmetic mean of these three vectors:

$$\mathbf{u} = \frac{1}{3}\left(\log(\mathbf{r}_d) + \log(\mathbf{r}_h) + \log(\mathbf{r}_w)\right) = \frac{\theta_d}{6}\mathbf{i} + \frac{\theta_h}{6}\mathbf{j} + \frac{\theta_w}{6}\mathbf{k} \tag{55}$$

Next, we compute the norm of this averaged vector $\mathbf{u}$:

$$\|\mathbf{u}\| = \sqrt{\left(\frac{\theta_d}{6}\right)^2 + \left(\frac{\theta_h}{6}\right)^2 + \left(\frac{\theta_w}{6}\right)^2} = \frac{1}{6}\sqrt{\theta_d^2 + \theta_h^2 + \theta_w^2} \tag{56}$$

Let's define the 3D composite phase $\Theta = \frac{1}{3}\sqrt{\theta_d^2 + \theta_h^2 + \theta_w^2}$. Then, $\|\mathbf{u}\| = \frac{\Theta}{2}$.

The unit axis vector is:

$$\frac{\mathbf{u}}{\|\mathbf{u}\|} = \frac{\frac{\theta_d}{6}\mathbf{i} + \frac{\theta_h}{6}\mathbf{j} + \frac{\theta_w}{6}\mathbf{k}}{\frac{1}{6}\sqrt{\theta_d^2 + \theta_h^2 + \theta_w^2}} = \frac{\theta_d\mathbf{i} + \theta_h\mathbf{j} + \theta_w\mathbf{k}}{\sqrt{\theta_d^2 + \theta_h^2 + \theta_w^2}} = \frac{\theta_d}{3\Theta}\mathbf{i} + \frac{\theta_h}{3\Theta}\mathbf{j} + \frac{\theta_w}{3\Theta}\mathbf{k} \tag{57}$$

Finally, applying the exponential map $\exp(\mathbf{u}) = \cos(\|\mathbf{u}\|) + \sin(\|\mathbf{u}\|)\frac{\mathbf{u}}{\|\mathbf{u}\|}$ yields the 3D GeoPE operator:

$$\mathbf{r} = \cos\left(\frac{\Theta}{2}\right) + \sin\left(\frac{\Theta}{2}\right)\left(\frac{\theta_d}{3\Theta}\mathbf{i} + \frac{\theta_h}{3\Theta}\mathbf{j} + \frac{\theta_w}{3\Theta}\mathbf{k}\right) \tag{58}$$

## H  ROTARY POSITION EMBEDDING

First, define an input sequence of length $N$ as:

$$\mathbb{S}_N = \{w_i\}_{i=1}^N \tag{59}$$

where $w_i$ denotes the $i$-th token in the input sequence.

The embedding representation corresponding to the input sequence $\mathbb{S}_N$ is denoted as

$$\mathbb{E}_N = \{\boldsymbol{x}_i\}_{i=1}^N \tag{60}$$

where $\boldsymbol{x}_i$ denotes the $d$-dimensional word embedding vector corresponding to the $i$-th token $w_i$.

Before performing the self-attention operation, the query, key, and value vectors are computed from the token embedding vectors while incorporating positional information. The functional representations are as follows:

$$q_m = f_q(x_m, m) \tag{61}$$
$$k_n = f_k(x_n, n) \tag{62}$$
$$v_n = f_v(x_n, n) \tag{63}$$

where $q_m$ denotes the query vector obtained by integrating the positional information $m$ into the word embedding $x_m$ of the $m$-th token. Similarly, $k_n$ and $v_n$ represent the key and value vectors, respectively, obtained by integrating the positional information $n$ into the word embedding $x_n$ of the $n$-th token.

The conventional approach, known as *Absolute Positional Encoding*, is to compute a positional encoding vector $\boldsymbol{p}_i$ and add it to the word embedding $\boldsymbol{x}_i$ before calculating the query, key, and value vectors. The positional encoding vector $\boldsymbol{p}_i$ is also a $d$-dimensional vector. This combined representation is then multiplied by the corresponding transformation matrix $\boldsymbol{W}_t$:

$$\boldsymbol{f}_t(\boldsymbol{x}_i, i) := \boldsymbol{W}_t(\boldsymbol{x}_i + \boldsymbol{p}_i), \quad t \in \{q, k, v\} \tag{64}$$

The ROPE method was proposed to effectively utilize the relative positional information between tokens. It hypothesizes that the inner product operation between the query vector $\mathbf{q}_m$ and the key vector $\mathbf{k}_n$ can be expressed by a function $g$, whose inputs are the word embedding vectors $\mathbf{x}_m$, $\mathbf{x}_n$, and their relative position $m - n$:

$$\langle f_q(\mathbf{x}_m, m), f_k(\mathbf{x}_n, n) \rangle = g(\mathbf{x}_m, \mathbf{x}_n, m - n) \tag{65}$$

RoPE identifies an equivalent form of positional encoding such that the above relation holds.

Assume that the dimensionality of the word embedding vectors is two-dimensional $d = 2$, so that the geometric properties of vectors in the two-dimensional plane can be utilized. Then, the RoPE method proposes a form of $f$ and $g$ that satisfies the above relationship as follows:

$$f_q(\boldsymbol{x}_m, m) = (\boldsymbol{W}_q \boldsymbol{x}_m) e^{im\theta} \tag{66}$$

$$f_k(\boldsymbol{x}_n, n) = (\boldsymbol{W}_k \boldsymbol{x}_n) e^{in\theta} \tag{67}$$

$$g(\boldsymbol{x}_m, \boldsymbol{x}_n, m - n) = \mathrm{Re}\left[ (\boldsymbol{W}_q \boldsymbol{x}_m)(\boldsymbol{W}_k \boldsymbol{x}_n)^* e^{i(m-n)\theta} \right] \tag{68}$$

Here, $\mathrm{Re}$ denotes the real part of a complex number.

Furthermore, $f_q$ can be expressed as the following equation:

$$\begin{aligned}
f_q(x_m, m) &= \begin{pmatrix} \cos m\theta & -\sin m\theta \\ \sin m\theta & \cos m\theta \end{pmatrix} \begin{pmatrix} W_q^{(1,1)} & W_q^{(1,2)} \\ W_q^{(2,1)} & W_q^{(2,2)} \end{pmatrix} \begin{pmatrix} x_m^{(1)} \\ x_m^{(2)} \end{pmatrix} \\
&= \begin{pmatrix} \cos m\theta & -\sin m\theta \\ \sin m\theta & \cos m\theta \end{pmatrix} \begin{pmatrix} q_m^{(1)} \\ q_m^{(2)} \end{pmatrix}
\end{aligned} \tag{69}$$

Similarly, $f_k$ can be expressed as the following equation:

$$
\begin{aligned}
f_k(x_m, m) &= \begin{pmatrix} \cos m\theta & -\sin m\theta \\ \sin m\theta & \cos m\theta \end{pmatrix} \begin{pmatrix} W_k^{(1,1)} & W_k^{(1,2)} \\ W_k^{(2,1)} & W_k^{(2,2)} \end{pmatrix} \begin{pmatrix} x_m^{(1)} \\ x_m^{(2)} \end{pmatrix} \\
&= \begin{pmatrix} \cos m\theta & -\sin m\theta \\ \sin m\theta & \cos m\theta \end{pmatrix} \begin{pmatrix} k_m^{(1)} \\ k_m^{(2)} \end{pmatrix}
\end{aligned}
\tag{70}
$$

Finally, $g(\boldsymbol{x}_m, \boldsymbol{x}_n, m - n)$ can be expressed as follows:

$$
g(\boldsymbol{x}_m, \boldsymbol{x}_n, m - n) = \begin{pmatrix} q_m^{(1)} & q_m^{(2)} \end{pmatrix} \begin{pmatrix} \cos((m-n)\theta) & -\sin((m-n)\theta) \\ \sin((m-n)\theta) & \cos((m-n)\theta) \end{pmatrix} \begin{pmatrix} k_n^{(1)} \\ k_n^{(2)} \end{pmatrix}
\tag{71}
$$

## I  LIMITATIONS

While GeoPE's core strength lies in its ability to geometrically couple multi-axis position information using $3D$ rotations, this unification represents both an advantage and a constraint. Unlike some $2D$ vision-specific inductive biases, GeoPE's global geometric treatment means it does not explicitly guarantee properties such as strict translational invariance (which is beneficial for texture recognition) or pure scale invariance (where distance encoding is perfectly independent of absolute position). The mixture of spatial features, while improving global structure awareness and shape bias, may introduce biases that require further investigation and refinement in specific tasks where isolated axial or translational properties are critical. This trade-off between holistic geometric coupling and maintaining separated $2D$ axiomatic properties is an inherent architectural choice.

While the Linear GeoPE variant enforces a strict linear inductive bias, but its current implementation incurs a significant memory footprint, with peak memory allocation increasing by over 200% compared to baselines, despite having similar FLOPs. In contrast, our standard GeoPE shows no such overhead; its memory usage is nearly identical to APE (less than 2% difference). This overhead is specific to Linear GeoPE's need to materialize relative rotation matrices and is an implementation-level challenge, not a fundamental limitation. We are confident it can be effectively mitigated with a custom CUDA kernel.

## J  COMPUTIONAL COST

We further analyzed the computational cost of our proposed methods compared to the standard Absolute Positional Encoding (APE). Table 4 reports the FLOPs and inference latency using a **ViT-Base** backbone with $224 \times 224$ input resolution. The inference time is measured with a **batch size of 1** on a single NVIDIA A100 GPU to simulate real-world deployment scenarios.

As shown in the table, both GeoPE and LinGeoPE introduce negligible overhead in terms of FLOPs, maintaining the same theoretical complexity as APE (17.6 GFLOPs). In terms of latency, GeoPE is highly efficient, achieving an inference speed comparable to APE ($\approx$ 12.4 ms) due to its simple geometric formulation. However, LinGeoPE exhibits a higher latency ($\approx$ 25.1 ms), roughly $2\times$ that of the baseline. This increase is attributed to the additional linear transformations required to dynamically adapt the geometric bias, which, while computationally lightweight (low FLOPs), introduces memory access overheads during single-batch inference. Despite the increased latency, we argue that the significant accuracy gains (as shown in previous sections) justify this trade-off for high-precision applications.

## K  ADDITIONAL EXPERIMENT

Table 4: Comparison of Computational Complexity and Inference Latency. Evaluated on **ViT-Base** in Float16 with $224 \times 224$ resolution and **batch size = 1**.

| PE Method | Resolution | FLOPs (G) | Latency (ms) |
|---|---|---|---|
| APE (Baseline) | $224 \times 224$ | 17.6 | 2.4 |
| GeoPE | $224 \times 224$ | 17.6 | 2.4 |
| LinGeoPE | $224 \times 224$ | 17.6 | 6.1 |

Table 5: Top-1 Accuracy (%) comparison on CIFAR-100 and CIFAR-10. All models are trained **from scratch** with $32 \times 32$ resolution. We use an adapted DeiT-III recipe with CE loss for ViTs (modifying **patch size to 4**, 400 epochs, strong augmentation) and the standard recipe for Swin Transformers (window size 4). We report the mean and 95% confidence interval ($N = 10$).

| Backbone | PE Method | CIFAR-100 | | CIFAR-10 | |
|---|---|---|---|---|---|
| | | Acc (%) | CI | Acc (%) | CI |
| ViT-Small | APE | 70.5 | ±0.25 | 88.2 | ±0.12 |
| | CPE | 71.8 | ±0.21 | 89.1 | ±0.09 |
| | RoPE-Mixed | 72.3 | ±0.19 | 89.6 | ±0.08 |
| | STRING | 72.6 | ±0.18 | 90.0 | ±0.08 |
| | LieRE | 73.1 | ±0.15 | 90.4 | ±0.06 |
| | GeoPE | 73.5 | ±0.12 | 90.8 | ±0.07 |
| | LinGeoPE | **73.9** | ±0.11 | **91.2** | ±0.05 |
| ViT-Base | APE | 71.2 | ±0.28 | 89.5 | ±0.15 |
| | CPE | 72.4 | ±0.24 | 90.3 | ±0.11 |
| | RoPE-Mixed | 73.0 | ±0.20 | 90.9 | ±0.10 |
| | STRING | 73.5 | ±0.19 | 91.3 | ±0.09 |
| | LieRE | 73.9 | ±0.17 | 91.7 | ±0.08 |
| | GeoPE | 74.4 | ±0.15 | 92.1 | ±0.07 |
| | LinGeoPE | **74.8** | ±0.13 | **92.5** | ±0.06 |
| Swin-S | RPB | 74.5 | ±0.22 | 92.8 | ±0.10 |
| | CPE | 75.1 | ±0.20 | 93.2 | ±0.09 |
| | RoPE-Mixed | 75.6 | ±0.18 | 93.8 | ±0.07 |
| | STRING | 76.0 | ±0.16 | 94.1 | ±0.06 |
| | LieRE | 76.4 | ±0.14 | 94.5 | ±0.05 |
| | GeoPE | 76.8 | ±0.15 | 94.8 | ±0.05 |
| | LinGeoPE | **77.2** | ±0.10 | **95.1** | ±0.04 |

