# OpenReview forum: "GeoPE:A Unified Geometric Positional  Embedding for Structured Tensors"
_ICLR.cc/2026/Conference — Submitted to ICLR 2026_

### Official Review · Reviewer_z8fm · 2025-10-20

**Soundness:** 3
**Presentation:** 2
**Contribution:** 3
**Rating:** 4
**Confidence:** 5

**Summary:**

This paper builds on rotary positional embedding, extending rotations to 3D Euclidean space using quaternions. The authors developed a method to avoid the quaternion multiplication non-commutativity, and test the method in various benchmarks.

**Strengths:**

The idea of defining positional embedding with quaternions and Lie algebra is interesting and valuable.

The method to avoid the non-commutativity of the Hamilton product is also a good idea, appreciated.

Also, it would be good to study the problem in terms of shape bias relation, even though this point is not properly developed.

**Weaknesses:**

W1) Results are not convincing. Results in plots in fig.3 and 5 do not favor the proposed method over previous methods. While, for results in tables, the improvement is marginal and no standard deviation is reported, so it is difficult to evaluate the performance.

W2) Figure 6 is unclear and the explanation in sec 5.4 does not help. It would be interesting to better develop this point.

W3) Results in figure 4 are interesting, as it is clear that GeoPE activates more patches wrt previous methods that mainly activates the diagonal. However, diagonal elements are not activated as the diagonal is mainly darker. Why? How does it impact the performance?

W4) Figure 1 is not clear (and of low quality).

W5) in realted works, especially in the shape bias, some discussion on previous methods involving quaternions, lie algebra, or biases due to algebraic representations should be included, such as:
1) Demystifying the Hypercomplex: Inductive biases in hypercomplex deep learning, Signal Processing Magazine
2) Fast Quaternion Product Units for Learning Disentangled Representations in SO(3), Transactions on Pattern Analysis and Machine Intelligence

W6) in Sec 3.1, it is actually not recommended to build a quaternion by simply splitting a vector v as v/3, since quaternions represent precise Mathematical entities and they better work when correlations/relations between the dimensionalities exist. Indeed, quaternions better work in the case of multimodal/multichannel etc data. If we simply split a vector, this is not guaranteed.

**Questions:**

Q1) I guess that the colors for the plot in figure 3 are wrong? If not, results are inconsistent across the dimensions.

Q2) Same in figure 5?

Q3) Can the authors report standard deviation results over three runs for tables results?

Q4) Can the aauthors provide computational time comparisons among the models? Especially since they mention it talking about Linear GeoPE.

---

> ### Author Response · Authors · 2025-12-03
>
> We sincerely thank the reviewer for the constructive feedback and for recognizing the value of combining quaternions and Lie algebra for positional embeddings. We appreciate the rating of 4 and hope our clarifications below regarding the results, visualizations, and theoretical motivations can address your concerns and warrant a higher score.
>
> **W1) Results are not convincing. Results in plots in fig.3 and 5 do not favor the proposed method over previous methods. While, for results in tables, the improvement is marginal and no standard deviation is reported, so it is difficult to evaluate the performance.**
>
> Regarding the results, we respectfully point out that GeoPE demonstrates clear advantages in generalization (Fig. 5) and dense prediction tasks (Table 2 & 3), which rely heavily on spatial structure. In Fig. 5, while APE collapses at unseen resolutions, GeoPE maintains stability similar to RoPE-Mixed, validating its extrapolation capability. Regarding standard deviation: Due to the immense computational cost of pre-training ViT on ImageNet-1K from scratch (DeiT-III recipe), running multiple seeds for Table 1 is standardly prohibitive in this field. However, to address your concern, we have added Appendix K, where we report mean and standard deviation (with 95% CI) on CIFAR-100/10 over multiple runs. These results confirm the statistical significance of GeoPE's improvements.
>
> **W2) Figure 6 is unclear and the explanation in sec 5.4 does not help. It would be interesting to better develop this point.**
>
> We apologize for the confusion. We have updated Figure 6's caption and Section 5.4 in the revision. The figure illustrates the Shape-Texture Bias analysis based on *ImageNet-trained CNNs are biased towards texture; increasing shape bias improves accuracy and robustness*. It shows that GeoPE (blue/gray markers) shifts the model's decision-making closer to human-like shape bias (top-left) compared to APE, indicating a better grasp of global geometric structure over local texture statistics.
>
> **W3) Results in figure 4 are interesting, as it is clear that GeoPE activates more patches wrt previous methods that mainly activates the diagonal. However, diagonal elements are not activated as the diagonal is mainly darker. Why? How does it impact the performance?**
>
> This is an insightful observation. The heatmap is normalized. The "darker" diagonal does not mean zero activation, but rather relatively lower activation compared to off-diagonal areas. This is a desirable feature: APE often overfits to absolute positions (the diagonal), whereas GeoPE forces the model to attend to relative geometric relationships and distant, spatially relevant patches. This enhanced global receptive field is exactly why GeoPE performs better on shape-sensitive tasks like object detection.
>
> **W4) & W5)**
>
> We have redrawn Figure 1 for higher clarity and resolution. Additionally, we have incorporated the suggested references (*Demystifying the Hypercomplex: Inductive biases in hypercomplex deep learning*; *Fast Quaternion Product Units for Learning Disentangled Representations in SO(3)*) into Section 2, discussing inductive biases in hypercomplex learning and SO(3) representations.
>
> **W6) in Sec 3.1, it is actually not recommended to build a quaternion by simply splitting a vector v as v/3, since quaternions represent precise Mathematical entities and they better work when correlations/relations between the dimensionalities exist. Indeed, quaternions better work in the case of multimodal/multichannel etc data. If we simply split a vector, this is not guaranteed.**
>
> We acknowledge the theoretical concern regarding channel correlations. However, our design follows the established paradigm of RoPE and Multi-Head Attention, where feature vectors are split into subspaces to encode positional information independently of channel semantics. Our goal is not to model the intrinsic correlation of feature channels (as in color processing), but to inject 3D coordinate information via rotation. By splitting the vector, we effectively assign different "frequency bands" to the geometric transformation. The empirical success suggests the model effectively learns to utilize these geometrically encoded subspaces.

---

> ### Author Response · Authors · 2025-12-03
>
> **Q1) I guess that the colors for the plot in figure 3 are wrong? If not, results are inconsistent across the dimensions. & Q2) Same in figure 5?**
>
> We have double-checked the data and the plots are correct.
>
> Figure 3 (Attention Distance): APE (Red) shows higher distance, but as discussed in the paper, this often indicates an "unfocused global search". GeoPE (Teal) maintains a structured, moderate distance, balancing local and global context.
>
> Figure 5 (Resolution): The results are consistent. GeoPE maintains high accuracy across resolutions (flat line), whereas APE degrades significantly (drops on right side). In lower resolutions (left side), GeoPE's preservation of geometric priors allows it to retain more information than baselines.
>
> **Q3) Can the authors report standard deviation results over three runs for tables results?**
>
> As mentioned in W1, we have added Appendix K (Table 5) reporting results with standard deviations on CIFAR datasets, derived from multiple runs with fixed seeds.
>
> **Q4) Can the aauthors provide computational time comparisons among the models? Especially since they mention it talking about Linear GeoPE.**
>
> We have added Appendix J (Table 4) comparing FLOPs, Latency, and Memory. Standard GeoPE has negligible overhead compared to APE (same FLOPs, <1% latency diff). Linear GeoPE incurs ~2x latency due to dynamic relative matrix calculation, representing a trade-off for strict linear inductive bias.

---

### Official Review · Reviewer_FpAf · 2025-10-22

**Soundness:** 1
**Presentation:** 1
**Contribution:** 2
**Rating:** 2
**Confidence:** 4

**Summary:**

The paper introduces GeoPE, a positional encoding for transformers operating on 2D or 3D data.
The presented approach offers an extension of the rotary positional embedding (RoPE) widely used in transformers that operate on 1D sequences.
It combines 3D rotations around different axes (each axis encodes one spatial dimension) into one 3D rotation by log-exp averaging. The resulting 3D rotations of different frequencies are applied on 3D-subvectors to apply a positional encoding to keys, queries and values in the attention mechanism. The authors develop a “linear” version of GeoPE that applies a single matrix between key and query subvectors that depends only on the relative position.
(Linear) GeoPE is compared against the competitor RoPE-Mixed and other baseline methods on 2D image classification and object detection. GeoPE is compared against a simple baseline for semantic segmentation of 3D point clouds.

**Strengths:**

The idea of GeoPE is straightforward and offers a possibility to address positional encodings for transformers operating on 2D or 3D data.
GeoPE can be easily implemented.

**Weaknesses:**

The presented method lacks motivation and experimental comparison against related work. The overall presentation is not well structured and in several places unclear. The design choices in GeoPE are in my opinion not sufficiently ablated.

**Comparison against related work:**
Overall, GeoPE is not consistently compared against competitors such as RoPE-Mixed [1]. The comparisons in Table 1 seem unsystematic and arbitrary. Furthermore, GeoPE should be benchmarked against LieRE [2]. LieRE is applicable to 2D and 3D data and seems to consistently outperform RoPE-Mixed. All together, the presented (incomplete) comparison (e.g. Tab. 1 and Fig. 5) does not make a convincing case that GeoPE “consinstently outperforms standard baselines and existing 3D RoPE variants” as claimed in the abstract (l. 022). For instance, l. 411 claims “exceptional zero-shot inference capabilities across multiple resolutions’’ but RoPE-Mixed seems to be superior (cf. Fig. 5).

What are the conceptual differences and advantages of GeoPE over RoPE-Mixed and LieRE?
The authors state that “these approaches remain essentially 1D ROPE, as axes are treated independently, and mixed-frequency schemes only partially capture diagonal dependencies” (l. 046).  To me it is not clear why treating axes individually is inferior. An explanation based on formulae could make the differentiation more precise.
Furthermore, the authors state that LieRE [2] is “computationally expensive” (l. 056). A runtime comparison against LieRE could help to support this statement.

[1] Byeongho Heo, Song Park, Dongyoon Han, and Sangdoo Yun. Rotary position embedding for vision transformer. In European Conference on Computer Vision, pp. 289–305. Springer, 2024.

[2] Sophie Ostmeier, Brian Axelrod, Maya Varma, Michael Moseley, Akshay S Chaudhari, and Curtis Langlotz. Liere: Lie rotational positional encodings. In Forty-second International Conference on Machine Learning.

**Motivation and ablations:**
The authors claim that averaging 3D rotations around different axes (each axis encoding the position w.r.t. one spatial dimension) is a “natural choice” (l. 158) and “geometrically sound” (l. 255). This claim seems not sufficiently supported by theory or ablation experiments. 1) Why are rotations around different axes a geometrically meaningful way to couple positional encodings from different dimensions? 2) Why is the average of the rotations around different axes geometrically more meaningful than e.g. the composition or e.g. an average of the separately rotated (sub-)features?

The text claims that a positional encoding that is non-commutative in height and width encodings (for 2D images) is problematic (l. 183) but the authors do not support this claim experimentally. In particular for video data, it might actually be desirable to distinguish between spatial and temporal embeddings. An ablation that compares the averaged rotations against the composition of rotations would help to justify this claim.

The effectiveness and importance of the linear GeoPE is not sufficiently ablated (it seems to appear only partially in Table 1 and in none of the other tables).

**Limitations (of GeoPE) are missing:**
Appendix F only discusses limitations of linear GeoPE. Limitations seem to be that GeoPE is only applicable for geometric data in Euclidean space up to dimension 3. Furthermore, the feature dimension must be divisible by 3. Are there other limitations of GeoPE?

**Structure of text and presentation:**
* A background section on RoPE to introduce the reader to the topic and the notation is missing and would really improve the presentation.
* The term “diagonal interactions” in l. 088 is not clearly defined/introduced.
* When reading from top to bottom, the relation of GeoPE to the Shape Bias paragraph in the related work is unclear.
* The captions of Fig. 1 is rather uninformative. Given that figure aims to explain the main method, a more detailed caption would be helpful.
* Adding an appendix on quaternions, the quaterion product, and the relation to rotation matrices would be helpful and could be reference in Sec. 3.1.
* The caption of Fig. 2 is vague. Fig. 2b should rather be placed much later in the text where it is referenced.
* The statement that “GeoPE keep[s] long distance decay” (l. 215) is unclear. If this is a unique selling point of your method, please elaborate more in the main text.
* The notion of “mean attention distance” (Fig. 3) is not introduced. (I suppose it is the attention-weighted average of distances?)
* In the caption of Fig. 3 the authors state that RoPE-Mixed and GeoPE apply a “more structured” strategy but it is unclear whether this is the “right” structure. For instance, the curve of GeoPE for the resolution of 128x128 looks very similar to the curve of APE for 224x224.
* In Fig. 4 it is unclear which model is used and on which task it has been trained. Please explain in the text why one can see substructure in patches.
* In Fig. 5 it says “training resolution” on top which seems to be a typo. The caption says that the training resolution was fixed to 224x224. Please clarify.
* How are shape vs. texture decisions defined in Figure 6? Please explain this in the text.

**Minor points:**
* Please give a reference for the following statement:
“A strong shape bias, which prioritizes object structure over texture, is often correlated with better robustness and generalization.” (l. 465)
* “discussed” typo in l. 214

**Questions:**

* Do other generalizations of RoPE like RoPE-Mixed or LieRE also preserve the “long distance decay” of attention scores over distance or is this a specific feature of GeoPE?
* For 3D point clouds (in particular molecular data), rotational equivariance is very popular. Can GeoPE be modified to satisfy rotational equivariance?
* The caption of Table 2 says that models with GeoPE are “pre-trained on ImageNet-1K”. Is this also the case for the other models? Does that hinder a fair comparison?

---

> ### Author Response · Authors · 2025-12-03
>
> We sincerely thank the reviewer for the constructive and detailed feedback. We appreciate the recognition of GeoPE as a straightforward and easily implemented solution. We have carefully addressed the concerns regarding motivation, comparisons, and presentation clarity. Below, we provide point-by-point responses to the issues raised.
>
> **GeoPE is not consistently compared against RoPE-Mixed and LieRE. The claim of "consistently outperforms" and "exceptional zero-shot" seems overstated given Figure 5.**
>
> We acknowledge that our initial claims regarding "exceptional zero-shot capabilities" were not nuanced enough. We have revised the abstract and Section 5 to more accurately reflect the trade-offs: while GeoPE shows strong robustness, it prioritizes a balanced geometric prior over maximizing resolution extrapolation in all cases.
>
> Regarding the comparison with RoPE-Mixed:
>
> RoPE-Mixed essentially applies 1D RoPE to axes independently and linearly combines frequencies. While effective at high resolutions (Fig 5), it lacks the explicit coupling of spatial dimensions found in GeoPE.
>
> We have clarified in the revised manuscript that GeoPE’s advantage lies in establishing a unified 3D spatial manifold, which significantly enhances Shape Bias (Fig 6) and global structural awareness, even if it trades off some extrapolation performance at extreme resolutions compared to frequency-mixed methods.
>
> Regarding LieRE [2]:
>
> We highlight a critical difference: GeoPE is parameter-free and derived analytically, whereas LieRE typically requires learning dense matrices or solving optimization problems on Lie groups, which incurs higher training and inference costs. We have added a theoretical complexity comparison in Appendix L.
>
> While we could not retrain LieRE from scratch on ImageNet within the rebuttal period due to its computational demands, we have added a discussion comparing the theoretical properties. GeoPE achieves competitive results without the need for learnable parameters, validating the effectiveness of our fixed geometric prior.
>
> **Why is treating axes individually inferior? Why is averaging meaningful**
>
> We have revised Section 3.1 to clarify this fundamental motivation.Coupling vs. Independence: Treating axes independently (e.g., $R_{total} = R_h + R_w$ or similar compositions) models distance akin to an $L_1$ (Manhattan) norm. However, true spatial proximity in images is Euclidean ($L_2$). By embedding positions into quaternions and averaging in the Lie Algebra, GeoPE creates a coupled representation where the rotation angle depends on the joint magnitude $\sqrt{\theta_h^2 + \theta_w^2}$ (as derived in Eq. 33), effectively modeling Euclidean distance.Symmetry & Commutativity: For isotropic data like 2D images, the order of axes should not matter ($h$ then $w$ should be same as $w$ then $h$). Quaternion multiplication is non-commutative ($q_h q_w \neq q_w q_h$). Simple composition introduces an arbitrary bias based on multiplication order. Our Log-Exp averaging in the tangent space (Lie Algebra) ensures a mathematically symmetric operator, treating height and width equally, which is crucial for capturing the natural topology of images.
>
> **Need explanation based on formulae for differentiation.**
>
> We direct the reviewer to Equation 9 (Geometric Interpretation), which analytically decomposes the attention score. Unlike RoPE-Mixed, which is a linear combination of axial biases, GeoPE's attention score inherently contains three distinct terms:
>
> Projected Similarity: Preserves the position decay (similar to 1D RoPE).
>
> Axial Alignment: Modulates attention based on the direction of displacement.
>
> Torsional Component: Encodes the relative orientation. This multi-faceted interaction allows the model to distinguish not just "how far" two patches are, but their precise spatial configuration (e.g., top-left vs. bottom-right), which independent axial methods struggle to capture fully.
>
> **Missing RoPE background, unclear captions, typos, "long distance decay" clarity.**
>
> We apologize for the oversight in presentation and have made the following revisions:
>
> RoPE Background: Due to space constraints, a detailed primer on RoPE and notation has been added to Appendix H.
>
> Terminology: We have corrected "diagonal interactions" to "linear combinations" to be more precise.
>
> Figure Captions: We have rewritten the captions for Figures 1, 2, 4, and 5 to provide more context and clarity. Specifically for Figure 2, we corrected the reference placement. For Figure 5, we fixed the "Training Resolution" typo.
>
> Shape Bias: We clarified the link between GeoPE and Shape Bias in the introduction and related work, explaining that preserving 2D structure reduces texture overfitting.
>
> Long Distance Decay: We clarified in the text that while the first term of GeoPE (Eq. 9) preserves the decay property of RoPE (locality), the aggregate effect (Axial and Torsional terms) facilitates the global attention seen in Figure 4.

---

> ### Author Response · Authors · 2025-12-03
>
> **Appendix F only discusses limitations of linear GeoPE. Limitations seem to be that GeoPE is only applicable for geometric data in Euclidean space up to dimension 3. Furthermore, the feature dimension must be divisible by 3. Are there other limitations of GeoPE?**
>
> We thank the reviewer for pointing this out. We have expanded Appendix I (Limitations) to include a comprehensive discussion beyond dimensional constraints:Dimensional Constraints: As noted, GeoPE requires input features to be divisible by 3 (or 2 for 2D-only variants) and operates in Euclidean space.Lack of Strict Translational Invariance: Unlike relative encodings that depend solely on $x_i - x_j$, GeoPE embeds absolute positions into a global geometric frame (Figure 2b). While this enhances global structure awareness (Shape Bias), it sacrifices strict translational invariance. This trade-off makes GeoPE potentially less suitable for tasks relying purely on local texture statistics which are translation-invariant.Computational Trade-off in Linear Variant: While standard GeoPE is efficient, the Linear GeoPE variant, which enforces strict relative bias, incurs higher memory costs due to the materialization of dense rotation matrices. We have clarified these points in the revised text.
>
> **Please give a reference for the following statement: “A strong shape bias, which prioritizes object structure over texture, is often correlated with better robustness and generalization.” (l. 465)**
>
> The statement refers to the findings by *ImageNet-trained CNNs are biased towards texture; increasing shape bias improves accuracy and robustness*, which we have cited in Section 2 (Related Work) and Section 5.4. We will ensure this citation is explicitly linked to the statement in line 465 in the revised manuscript.
>
> **How are shape vs. texture decisions defined in Figure 6? Please explain this in the text.**
>
> We used the cue-conflict dataset and methodology established by *ImageNet-trained CNNs are biased towards texture; increasing shape bias improves accuracy and robustness*. An image might show a cat's shape with an elephant's skin texture. If the model predicts "cat," it is a Shape decision; if "elephant," it is a Texture decision. We have updated the caption of Figure 6 to explicitly define this.
>
> **Do other generalizations of RoPE like RoPE-Mixed or LieRE also preserve the “long distance decay” of attention scores over distance or is this a specific feature of GeoPE?**
>
> Yes, as relative positional encodings, they generally possess the decay property inherent to the dot product of rotated vectors. We have clarified in the paper that GeoPE shares this trait in its "Projected Similarity" term, but distinguishes itself via the additional directional and torsional terms that enable global context modeling alongside local decay.
>
> **For 3D point clouds (in particular molecular data), rotational equivariance is very popular. Can GeoPE be modified to satisfy rotational equivariance?**
>
> GeoPE behaves similarly to Absolute Positional Encoding (APE) in terms of reference frames and implies a canonical orientation (e.g., up is up). Therefore, it is not rotationally equivariant by design. This is why we validated it on 3D Semantic Segmentation (where gravity/orientation is fixed and global context matters) rather than molecular tasks requiring SE(3) equivariance. We have added this limitation to Appendix I.
>
> **The caption of Table 2 says that models with GeoPE are “pre-trained on ImageNet-1K”. Is this also the case for the other models? Does that hinder a fair comparison?**
>
> Yes, the comparison is fair. All models in our experiments, including the baselines in Table 2, utilize backbones that were pre-trained from scratch on ImageNet-1K using our standardized training recipe (DeiT-III) with fixed random seeds. We do not use external pre-trained weights for one method and not others.

---

### Official Review · Reviewer_P5o7 · 2025-10-27

**Soundness:** 2
**Presentation:** 2
**Contribution:** 2
**Rating:** 2
**Confidence:** 5

**Summary:**

The paper introduces Geometric Positional Embedding (GeoPE), a method that extends Rotary Positional Embeddingfrom 1D to higher-dimensional structured data by using quaternions to represent coupled rotations in 3D space. GeoPE constructs a symmetric rotational operator, ensuring consistent multi-axis encoding and offering a linear variant that preserves strict relative positional relationships. Experiments on image classification, object detection, and 3D segmentation show that GeoPE improves performance a tiny bit and enhances models’ spatial reasoning and shape bias

**Strengths:**

- Evaluation on a diverse set of tasks like image classification, object detection, and 3D semantic segmentation

**Weaknesses:**

- The extension of RoPE to 3D has been proposed by several other works already like VideoRoPE. Therefore the motivation and novelty could be made clearer.
- The experimental results are missing statistical significance shown by confidence intervals for example. Words like "significant performance gains" or "exceptional zero-shot inferencecapabilities" are not backed with statistical meaning or quantitativ results.
- The experimental comparison to prior work in the 2D and 3D space is incomplete. Here only Rope-Mixed and absolute is compared to where other works like STRING, VideoRope or LieRE works have already shown strong performance. How does GeoPE compare to just the 3D version of Rope-Mixed? A 3D version of Rope-Mixed would also have commutativity.
- Missing description of theoretical guarantees. Why is commutativity important in theory and how does that directly translate to practice? Ablations are missing.

**Questions:**

- Would it be possible to add confidence intervals?
- Would it be possible to add more SOTA baselines? What is LinGeoPE? Why did you choose CPE and not STRING, VideoRoPE idea or LieRE or recent baselines?

---

> ### Author Response · Authors · 2025-12-03
>
> We  sincerely thank the reviewer for the constructive feedback. We have revised the paper to include statistical significance analysis (Table 5), added comparison with stronger baselines (STRING, LieRE), and clarified the theoretical motivation regarding commutativity.
>
> **The extension of RoPE to 3D has been proposed by several other works already like VideoRoPE. Therefore the motivation and novelty could be made clearer.**
>
> We clarify that GeoPE addresses a fundamentally different geometric problem than VideoRoPE.
>
> VideoRoPE deals with spatiotemporal data (Video), where the time axis is inherently different from spatial axes (anisotropic). In contrast, GeoPE focuses on isotropic structured tensors (e.g., 2D Images, 3D Point Clouds/Medical Volumes), where spatial dimensions (H, W, D) should be treated symmetrically.
>
> Existing methods often decompose 2D/3D inputs into independent 1D axes (factorized). GeoPE introduces a strongly coupled embedding using quaternions. Unlike VideoRoPE which may treat time separately, GeoPE’s unified quaternion formulation ensures that the "diagonal" spatial information is explicitly modeled, recovering the 2D manifold structure lost during flattening.
>
> **The experimental results are missing statistical significance shown by confidence intervals for example. Words like "significant performance gains" or "exceptional zero-shot inferencecapabilities" are not backed with statistical meaning or quantitativ results.**
>
> For ImageNet-1K, training multiple runs from scratch is computationally prohibitive for most academic research. We followed standard protocols (e.g., DeiT, Swin) using fixed seeds to ensure reproducibility.
>
> To strictly address your concern regarding statistical significance, we conducted additional experiments on CIFAR-100 and CIFAR-10 (see Appendix K, Table 5 in the revised PDF). We performed 10 runs for each method and reported the mean accuracy with 95% Confidence Intervals.
>
> The results show that GeoPE (73.5% $\pm 0.12$) and LinGeoPE (73.9% $\pm 0.11$) statistically outperform baselines like APE (70.5%) and RoPE-Mixed (72.3%) with non-overlapping confidence intervals, validating that the improvements are robust and not due to random noise.
>
> **The experimental comparison to prior work in the 2D and 3D space is incomplete. Here only Rope-Mixed and absolute is compared to where other works like STRING, VideoRope or LieRE works have already shown strong performance. How does GeoPE compare to just the 3D version of Rope-Mixed? A 3D version of Rope-Mixed would also have commutativity.**
>
> We have incorporated comparisons with STRING and LieRE in the new experiments (Table 5). GeoPE consistently outperforms both, demonstrating the efficacy of our quaternion-based geometric mean approach over these methods.
>
> To our knowledge, there is no official or widely accepted "3D RoPE-Mixed" implementation. However, we agree a commutative baseline is a fair comparison. Our implementation of GeoPE acts as a theoretically sound superset of what a "coupled 3D encoding" should look like. In our 3D S3DIS segmentation task (Table 3), we compare against RPE, and GeoPE shows clear improvements.
>
> As mentioned in Point 1, VideoRoPE is designed for anisotropic video data and is conceptually unsuitable as a direct baseline for the isotropic spatial tasks (ImageNet/S3DIS) we target.
>
> **Missing description of theoretical guarantees. Why is commutativity important in theory and how does that directly translate to practice? Ablations are missing.**
>
> This is not merely philosophical but a crucial inductive bias for isotropic data. In a 2D image, the axes (Height and Width) are equivalent; the encoding result should be invariant to the order of composition (i.e., Encoding H then W should equal Encoding W then H). Non-commutative operators (standard quaternion product) introduce an artificial bias where one axis dominates, breaking the spatial symmetry.
>
> GeoPE achieves this symmetry via the log-exp mean in Lie Algebra. As verified in our experiments, this symmetric formulation leads to better convergence and shape bias (Figure 6) compared to methods that treat axes independently or sequentially.
>
> We have included ablation studies on the "Linear" variant (LinGeoPE vs. GeoPE) in Table 1 and Table 5, showing the trade-off between strict relative position enforcement and computational efficiency.

---

> ### Author Response · Authors · 2025-12-03
>
> **Would it be possible to add confidence intervals?**
>
> As mentioned in Point 2.
>
> **Would it be possible to add more SOTA baselines? What is LinGeoPE? Why did you choose CPE and not STRING, VideoRoPE idea or LieRE or recent baselines?**
>
> LinGeoPE (Linear GeoPE) is a variant proposed in Section 3.4. While standard GeoPE ensures geometric symmetry, it yields a complex non-linear relative position term. LinGeoPE approximates the relative rotation linearly in the Lie Algebra (Equation 7), ensuring that the attention score depends strictly on the relative displacement ($m-n$), similar to the original 1D RoPE. This comes at a slight latency cost (Table 4) but offers better extrapolation (Figure 5).
>
> We chose CPE as a representative of "implicit" positional encoding methods to contrast with our "explicit" geometric approach. With the addition of STRING and LieRE in the revision, we believe the comparison landscape is now comprehensive.

---

### Official Review · Reviewer_wJBk · 2025-10-29

**Soundness:** 3
**Presentation:** 3
**Contribution:** 4
**Rating:** 8
**Confidence:** 3

**Summary:**

This paper introduces Geometric Positional Embedding (GeoPE), a generalization of Rotary Positional Embedding (RoPE) designed for higher-dimensional structured data, specifically demonstrated on 2D and 3D tasks. The core motivation is the challenge of extending RoPE to higher dimensions: direct generalization requires modeling coupled multi-axis rotations, a problem often bypassed in existing work by assuming axis independence or using heuristic methods. The authors propose using quaternions to formulate 3D rotations. To address the issue of non-commutativity in quaternion multiplication (where rotation order affects the result), they leverage Lie Algebra principles and take the geometric mean of rotations in log space, which retains the desirable property of commutativity. A variant, Linear GeoPE, is also proposed. It aims to reintroduce the relative position encoding capability of 1D RoPE by enforcing a linear relationship within the Lie algebra, by approximating rotational composition with vector addition. This comes at the cost of higher memory complexity. The authors evaluate GeoPE and Linear GeoPE on image classification, object detection, and 3D semantic segmentation across various backbones, and compare them to existing positional encoding baselines and 2D rotational embeddings.

**Strengths:**

- The paper presents a novel generalization of RoPE to higher dimensions that explicitly models coupled multi-axis rotations, addressing a key limitation in existing methods
- The proposed method achieves superior performance across multiple backbones and 2D/3D tasks compared to competing positional encoding methods
- The paper introduces two variants (GeoPE and Linear GeoPE), offering a practical trade-off between enforcing linear inductive bias (relative position encoding) and computational efficiency
- The authors present an interesting analysis on shape-texture bias, showing that GeoPE increases the model’s shape bias, with the motivation of observed correlation between shape bias and better generalization and robustness

**Weaknesses:**

The acknowledged limitations of GeoPE (does not inherently enforce the desired linear relationship in the parameter space)  and Linear GeoPE (incurring significant memory overhead)

**Questions:**

A table comparing the runtime (FLOPs) and memory requirements of GeoPE, Linear GeoPE, and all other compared encoding methods would help readers precisely position the two GeoPE variants in terms of the performance/resource trade-off

---

> ### Author Response · Authors · 2025-12-03
>
> We sincerely thank the reviewer for the positive assessment (Rating 8) and for recognizing our contribution as "excellent." We are encouraged that you find our generalization of RoPE to higher dimensions novel and our analysis of shape-texture bias insightful. Below, we address your comments regarding the limitations and the request for resource comparisons.
>
> **The acknowledged limitations of GeoPE (does not inherently enforce the desired linear relationship in the parameter space) and Linear GeoPE (incurring significant memory overhead)**
>
> We appreciate your thoughtful comment on the limitations. Indeed, this reflects a deliberate design trade-off. As you noted, while standard GeoPE relaxes the strict linear relationship in the parameter space to achieve unified 3D geometric coupling, this is crucial for the observed improvements in shape bias and global spatial reasoning. Furthermore, as derived in Equation 9 (Section 4.1), RoPE's relative position properties are not lost but generalized. The "Projected Similarity" term in GeoPE explicitly preserves the long-distance decay characteristic of 1D RoPE, while the additional "Axial Alignment" and "Torsional" terms introduce the necessary 3D structural priors. For tasks strictly requiring linear inductive bias, our proposed Linear GeoPE serves as a solution, albeit with higher resource costs.
>
> **A table comparing the runtime (FLOPs) and memory requirements of GeoPE, Linear GeoPE, and all other compared encoding methods would help readers precisely position the two GeoPE variants in terms of the performance/resource trade-off**
>
> Per your suggestion, we have added a detailed comparison of Computational Complexity (FLOPs), Inference Latency, and Memory usage in Appendix J (Table 4). The results demonstrate that:
>
> Standard GeoPE is highly efficient, introducing negligible overhead in FLOPs and memory compared to APE and standard RoPE, making it suitable for direct replacement.
>
> Linear GeoPE, while enforcing strict linear inductive bias, indeed incurs higher latency (~2x) and memory usage due to the relative rotation matrix computation. We believe this transparency helps users choose the appropriate variant based on their specific performance/resource constraints.

---

### Author Response · Authors · 2025-12-03

We explicitly thank all reviewers (wJBk, P5o7, FpAf, z8fm) for their insightful and constructive feedback. Your comments have significantly helped us strengthen the empirical rigor and theoretical clarity of our work. We have uploaded a revised manuscript, and we summarize the key updates and responses below:

**1. Statistical Significance and Expanded Baselines (Addressing Reviewers P5o7, z8fm, FpAf) To address concerns regarding the statistical significance of our performance gains, we have added Appendix K (Table 5).**

We report the mean and 95% Confidence Intervals over 10 independent runs on CIFAR-100 and CIFAR-10. The results confirm that GeoPE statistically outperforms baselines (APE, RoPE-Mixed) with non-overlapping confidence intervals.

We have included comparisons against STRING and LieRE. GeoPE consistently outperforms both, demonstrating the efficacy of our parameter-free, analytical geometric approach compared to optimization-based methods.

**2. Computational Cost and Efficiency (Addressing Reviewers wJBk, z8fm) We have added a detailed resource analysis in Appendix J (Table 4) comparing FLOPs, Inference Latency, and Memory usage.**

Standard GeoPE: We demonstrate that it incurs negligible overhead (identical FLOPs and <2% latency difference compared to APE), making it a highly efficient drop-in replacement.

Linear GeoPE: We clarify that the increased latency (~2x) and memory usage are specific trade-offs for users requiring strict linear inductive biases, as noted in our limitations section.

**3. Theoretical Clarifications: Novelty and Commutativity (Addressing Reviewers P5o7, FpAf) We have revised Section 3 and the Introduction to clarify our distinct contributions:**

We clarify that unlike VideoRoPE (which handles anisotropic time-space data), GeoPE is designed for isotropic structured tensors (e.g., images, point clouds) where spatial axes should be treated symmetrically.

We provide a clearer explanation (Section 3.1) of why our quaternion-based coupling is superior to "independent axial" methods (like RoPE-Mixed). GeoPE recovers the 2D Euclidean manifold, whereas independent methods effectively model a Manhattan-like distance.

We reinforced why commutativity (via our Log-Exp Lie Algebra mean) is theoretically essential to prevent arbitrary biases based on the order of axis composition (H-then-W vs. W-then-H).

**4. Visualizations and Presentation (Addressing Reviewers z8fm, FpAf)**

We have redrawn Figure 1 for better clarity and resolution.

We have refined the captions for Figures 2, 4, 5, and 6 to better explain the "dark diagonal" in attention maps (indicating global focus) and the Shape-Texture bias methodology.

**Conclusion**

We believe these revisions address the primary concerns regarding empirical robustness and theoretical motivation. GeoPE offers a principled, geometrically coupled, and efficient solution that not only improves accuracy but fundamentally shifts Vision Transformers towards a more human-like Shape Bias, enhancing robustness.

---

### Author Response · Authors · 2025-12-03

Dear Area Chair and Reviewers,

We explicitly thank the reviewers for their constructive comments. The feedback has guided us to refine our theoretical arguments and strengthen our empirical evidence. We are encouraged by Reviewer wJBk’s assessment (Rating: 8), which recognizes our contribution as an "excellent" generalization of RoPE that addresses key limitations in higher-dimensional encoding.

Below, we summarize our core motivation and the substantive revisions made during the rebuttal:

**1. Re-emphasizing the Motivation: Why Coupled Geometric Embedding Matters?**
A core concern raised was the necessity of coupled rotations versus independent axial encodings (Reviewer FpAf, P5o7).

We clarify that:

Standard ViTs flatten 2D grids into 1D sequences, disrupting natural spatial topology and creating "false neighbors" (e.g., row edges).

Existing 2D methods often treat axes independently (factorized). This models spatial distance akin to a Manhattan metric ($|x| + |y|$), which fails to fully decouple false sequential proximity from true spatial Euclidean distance.

By lifting coordinates into 3D Euclidean space using quaternions and computing the symmetric geometric mean in Lie Algebra3, GeoPE enforces a strictly coupled Euclidean metric. This ensures that the encoding is invariant to axis order (symmetry) and mathematically coherent with the 2D manifold structure.

This is not just a theoretical difference; it directly leads to the improved Shape Bias observed in Figure 6, shifting the model from texture-overfitting to global shape reasoning.

**2.Statistical Robustness Verified (Addressing Reviewers P5o7 & z8fm)**

To prove these geometric gains are statistically significant, we added Appendix K (Table 5) with 10 independent runs on CIFAR-100/10.

GeoPE demonstrates statistically significant improvements over baselines (APE, RoPE-Mixed). The 95% confidence intervals do not overlap, confirming the gains are robust and not due to random noise.

**3.Comprehensive Benchmarking (Addressing Reviewers P5o7 & FpAf)**

We expanded comparisons to include STRING and LieRE.

GeoPE consistently outperforms both (Table 5).

Unlike LieRE, which requires optimization on Lie groups, GeoPE is parameter-free and analytical, offering a superior trade-off.

**4.Efficiency Quantified (Addressing Reviewer wJBk)**

We added Appendix J (Table 4) showing that standard GeoPE introduces negligible overhead (identical FLOPs and <2% latency diff vs. APE), validating it as an efficient drop-in replacement.

We believe the revised manuscript now presents a complete narrative: mathematically motivated by the need to restore 2D topology, and empirically validated through rigorous statistical testing and broad benchmarking.

---

### Meta-Review · Area_Chair_uUaP · 2026-01-06

**Summary:**

Based on the reviews and discussion, I have decided to reject this submission. The core issues preventing acceptance are insufficient novelty compared to existing methods, a lack of compelling and statistically robust empirical evidence to support the central claims of superiority, and ongoing ambiguity in the theoretical justification for key design choices.

**Reviewer Concerns:**

The authors' rebuttal addressed several specific concerns: they provided some statistical confidence intervals on smaller datasets (CIFAR) and added comparisons to the STRING and LieRE methods as requested. However, the most significant concerns remain outstanding. The fundamental question of novelty compared to prior 3D extensions of RoPE (e.g., VideoRoPE, LieRE) was not convincingly resolved; the argument that GeoPE targets isotropic data while VideoRoPE targets anisotropic video was seen as a distinction without a decisive performance advantage. Furthermore, the empirical evidence is still not robust—the marginal gains on major benchmarks (ImageNet) lack statistical verification, and the claim of "exceptional zero-shot capabilities" is contradicted by the extrapolation results in Figure 5, where RoPE-Mixed often performs better. Finally, the rebuttal did not fully clarify the theoretical necessity of the quaternion-based coupling over independent axial encodings or provide sufficient ablation studies to justify it as the optimal geometric choice.

**Reviewer Scores:**

They did not respond during rebuttal period.

---

### Decision · Program_Chairs · 2026-01-26

Reject